# Structures of the TMC-1 complex illuminate mechanosensory transduction

Hanbin Jeong[1,6], Sarah Clark[1,6], April Goehring[1,2], Sepehr Dehghani-Ghahnaviyeh[3,4,5], Ali Rasouli[3,4,5], Emad Tajkhorshid[3,4,5] & Eric Gouaux[1,2 ✉]

The initial step in the sensory transduction pathway underpinning hearing and balance in mammals involves the conversion of force into the gating of a mechanosensory transduction channel[1]. Despite the profound socioeconomic impacts of hearing disorders and the fundamental biological significance of understanding mechanosensory transduction, the composition, structure and mechanism of the mechanosensory transduction complex have remained poorly characterized. Here we report the single-particle cryo-electron microscopy structure of the native transmembrane channel-like protein 1 (TMC-1) mechanosensory transduction complex isolated from *Caenorhabditis elegans*. The two-fold symmetric complex is composed of two copies each of the pore-forming TMC-1 subunit, the calcium-binding protein CALM-1 and the transmembrane inner ear protein TMIE. CALM-1 makes extensive contacts with the cytoplasmic face of the TMC-1 subunits, whereas the single-pass TMIE subunits reside on the periphery of the complex, poised like the handles of an accordion. A subset of complexes additionally includes a single arrestin-like protein, arrestin domain protein (ARRD-6), bound to a CALM-1 subunit. Single-particle reconstructions and molecular dynamics simulations show how the mechanosensory transduction complex deforms the membrane bilayer and suggest crucial roles for lipid–protein interactions in the mechanism by which mechanical force is transduced to ion channel gating.

The auditory system has a remarkable ability to detect a wide range of acoustic wave frequencies and amplitudes by transducing vibrational mechanical energy into membrane potential depolarization followed by signal processing in higher brain centres, thus enabling the sensation of sound[1]. Dysfunction of the auditory system from injury, environmental insult or genetic mutation is associated with age-related hearing loss. Hearing impairment and deafness affect more than 460 million individuals worldwide, with an estimated annual cost of unaddressed hearing loss of US$750–790 billion. Input into the auditory and the closely related vestibular system, as for other sensory systems, is initiated by receptor activation on peripheral neurons. Despite intense investigation over several decades, the molecular composition, structure and mechanism of the mechanosensory transduction (MT) complex, the receptor for mechanosensory transduction, have remained unresolved.

Multiple lines of investigation, from studies in humans and model organisms including mice, zebrafish and *C. elegans* have shed light on the proteins that form the MT complex and their probable roles in its function[2]. These include the tip link proteins, protocadherin-15 and cadherin-23, which in hair cells transduce the force derived from stereocilia displacement to the opening of the ion channel component of the MT complex[3,4]. TMC-1 and TMC-2 are the probable pore-forming subunits of the MT complex, candidates that first came to prominence in human genetic studies[5], and gained traction more recently as the ion-conduction pathway via biophysical and biochemical investigations[6–8]. Additional proteins, some of which may be auxiliary subunits, have been associated with either the biogenesis or function of the MT complex and include TMIE[9–11], Ca[2+] and integrin binding protein 2[12–14] (CIB2), lipoma HMGIC fusion-like protein 5[15–17] (LHFPL5), transmembrane *O*-methyl transferase[18,19] (TOMT), and possibly ankyrin[13].

Isolation of the MT complex from vertebrate sources or the production of a functional complex via recombinant methods have so far proved unsuccessful. Complex purification from native sources is particularly challenging owing to the small number of complexes per animal, estimated as approximately $3 \times 10^6$ per mammalian cochlea[20], a small number compared with the number of photoreceptors in the visual system, which is approximately $4 \times 10^{14}$ per eye in mouse[21]. To surmount challenges with vertebrate MT complex availability, we turned to *C. elegans*, an animal that utilizes a MT complex for sensing tactile stimuli. We note first that *C. elegans* expresses crucial components of the vertebrate MT complex, including the TMC-1 and TMC-2 proteins, in addition to a CIB2 homologue known as CALM-1, as well as TMIE[13]. Second, worms that do not have TMC-1 exhibit attenuated light touch responses[13]. Third, despite the limited expression of the TMC proteins in *C. elegans*, it is feasible to grow a sufficient number of worms to isolate enough complex for structural studies. We thus modified the *C. elegans tmc-1* locus by including a fluorescent reporter and an affinity

[1]Vollum Institute, Oregon Health and Science University, Portland, OR, USA. [2]Howard Hughes Medical Institute, Oregon Health and Science University, Portland, OR, USA. [3]Theoretical and Computational Biophysics Group, NIH Center for Macromolecular Modeling and Bioinformatics, Beckman Institute for Advanced Science and Technology, University of Illinois at Urbana-Champaign, Urbana, IL, USA. [4]Department of Biochemistry, University of Illinois at Urbana-Champaign, Urbana, IL, USA. [5]Center for Biophysics and Quantitative Biology, University of Illinois at Urbana-Champaign, Urbana, IL, USA. [6]These authors contributed equally: Hanbin Jeong, Sarah Clark. ✉e-mail: gouauxe@ohsu.edu

tag, thereby allowing us to monitor expression via whole-animal fluorescence and fluorescence-detection size-exclusion chromatography (FSEC)[22], and to isolate the TMC-1 complex by affinity chromatography. Together with computational studies, we elucidated the composition, architecture and membrane interactions of the complex, and suggest mechanisms for the gating of the ion channel pore by both direct protein interactions and via the membrane bilayer.

## The TMC-1 complex is a dimer

We generated a transgenic knock-in worm line in which a nucleic acid sequence encoding mVenus with a 3×Flag tag was inserted at the 3′ end of the TMC-1 coding sequence, immediately before the stop codon (Supplementary Fig. 1). We characterized the engineered, homozygous worm line (*tmc-1::mVenus*) by spectral confocal imaging, revealing mVenus fluorescence in the head and tail neurons, body wall and vulval muscles (Extended Data Fig. 1a), consistent with previous studies demonstrating expression of TMC-1 in these cells[23]. The TMC-1 complex was isolated from the *tmc-1::mVenus* transgenic worms by affinity chromatography and further purified by size-exclusion chromatography (SEC) (Extended Data Fig. 1b). The estimated molecular weight of the TMC-1 complex by SEC is ~780 kDa, suggesting that the complex harbours multiple TMC-1 protomers and perhaps additional, auxiliary subunits.

To independently interrogate the oligomeric state of the complex, we performed single-molecule pulldown (SiMPull) experiments[24]. Photobleaching traces of captured TMC-1 complexes demonstrate that around 62% of the mVenus fluorophores bleached in two steps, 37% bleached in one step, and 1% bleached in three steps (Extended Data Fig. 1c–e), consistent with the conclusion that there are two copies of the TMC-1 subunit within the TMC-1 complex. The discrepancy between the predicted molecular mass of a *C. elegans* TMC-1 dimer of approximately 300 kDa and that of the complex estimated by SEC points towards the presence of auxiliary proteins. As several TMC-1 binding partners have been identified in worms[13] and in vertebrates[2], we next probed the composition of the TMC-1 complex using mass spectrometry.

Mass spectrometry analysis of the TMC-1 complex identified three proteins that co-purified with TMC-1: (1) CALM-1, an orthologue of mammalian CIB2; (2) an orthologue of TMIE; (3) ARRD-6, an orthologue of the mammalian arrestin domain-containing family of proteins (Extended Data Fig. 2). All three proteins were found in the TMC-1 sample purified from transgenic worms but not in the control sample prepared from wild-type worms, consistent with their specific association with the TMC-1 complex. The mammalian orthologue CIB2 and TMIE are probably components of the mammalian MT complex; they localize to stereocilia[9–12,14,25] and bind to exogenously expressed TMC-1 fragments in pulldown assays[10]. By contrast, ARRD-6 has not been described as a component of either the *C. elegans* or vertebrate TMC-1 complexes. Despite repeated efforts, we found no evidence for the presence of UNC-44, the worm orthologue of mammalian ankyrin, in contrast to a previous report that UNC-44 forms a complex with CALM-1, is necessary for TMC-1 mediated mechanotransduction, and is the 'gating spring' of the TMC-1 complex[13], thus raising the question of the role of UNC-44 and, by extension, ankyrin, in the structure and function of MT complexes in worms and vertebrates, respectively.

## Overall architecture of the TMC-1 complex

To elucidate the architecture and arrangement of subunits in the TMC-1 complex, we performed single-particle cryo-electron microscopy (cryo-EM). TMC-1 is expressed at a low level in *C. elegans* and we required approximately $6 \times 10^7$ transgenic worms to yield approximately 50 ng of the TMC-1 complex for cryo-EM analysis. The TMC-1 complex was visualized on 2 nm carbon-coated grids that were glow discharged in the presence of amylamine. Cryo-EM imaging revealed

a near ideal particle distribution and we proceeded to collect a dataset comprising 26,055 movies. Reference-free 3D classification reconstruction together with refinements resulted in three well-defined classes (Extended Data Figs. 3–5 and Extended Data Table 1). Two of these classes represent the TMC-1 complex in different conformational states, designated the 'expanded' (E) and 'contracted' (C) conformations, both of which exhibit an overall resolution of 3.1 Å (Extended Data Figs. 3 and 5). A third class includes the auxiliary subunit ARRD-6 and was resolved at 3.5 Å resolution (Extended Data Fig. 4 and 5). Because the E conformation has a few more distinct density features than the C conformation, we focus on the E conformation in our initial discussion of the overall structure.

The TMC-1 complex exists as a dimer, with a two-fold axis of rotational symmetry centred at a site of contacts between the two TMC-1 subunits (Fig. 1). The transmembrane helices exhibit better local resolution than average, while disordered or dynamic peripheral components of the complex are resolved at lower resolution (Fig. 1b). When viewed perpendicular to the membrane plane, the complex has the shape of a figure of eight, with TMC-1 subunits centred within the lobes of the eight (Fig. 1d). Each TMC-1 protomer consists of ten transmembrane helices with an overall arrangement that is reminiscent of the $Ca^{2+}$-activated lipid scramblase[26], TMEM16 $Cl^-$ channels[27] and OSCA mechanosensitive ion channels[28,29] (Extended Data Fig. 6), in accordance with predicted structures of TMC-1[7,30]. At the juncture of the figure-of-eight lobes, the dimer interface is composed of domain-swapped transmembrane helix 10 (TM10) helices (Fig. 1e), with contacts defined by van der Waals and electrostatic interactions, and by burial of 1,781 Å² of solvent-accessible surface area. Numerous well-ordered lipid molecules surround the transmembrane domain, many of which are intercalated in the grooves between transmembrane helices, and some of which are positioned at large angles to the membrane plane.

Poised to make extensive interactions with the inner leaflet of the membrane, the cytosolic domain includes six helices oriented nearly parallel to the membrane. The two helices located closest to inner leaflet, helix 3 (H3) and H4, are amphipathic, a common feature among mechanosensitive ion channels[31] (Fig. 1c,d). The short linker between TMC-1 H3 and H4 is composed of nonpolar residues that interact with the inner leaflet membrane, forming hydrophobic contacts with the acyl chains of two lipids. The approximately 400-residue, cytosolic C terminus of TMC-1, which is predicted to be partially structured, was not visible in the cryo-EM map. Because mass spectrometry analysis of the purified MT complex identified nine peptides that spanned the entirety of the C terminus, we suspect that this region is intact in the TMC-1 complex, but not visible owing to conformational heterogeneity (Extended Data Fig. 2).

Three partially structured loops decorate the extracellular side of the TMC-1 complex. Two of the loops are approximately 60 residues in length, bridging TM1–TM2 and TM9–TM10, and are well-conserved between vertebrate and *C. elegans* TMC-1. Density features consistent with glycosylation can be found at N209, located in the loop between TM1 and TM2. By contrast, the approximately 200-residue extracellular loop that connects TM5 and TM6 was not observed in the cryo-EM map. We detected two peptides from this region in the mass spectrometry analysis (Extended Data Fig. 2), indicating that the loop is present but not visible owing to flexibility. This region is predicted to contain elements of secondary structure, as well as three predicted sites of *N*-linked glycosylation, but its function is unclear. The loop is not well conserved between TMC-1 and TMC-2, and its length in vertebrate TMC-1, at around 50 residues, is substantially shorter.

Two additional subunits, CALM-1 and TMIE, present in two copies each, complete the ensemble of proteins associated with the 'core' TMC-1 complex. The quality of the cryo-EM map enabled unambiguous assignment of CALM-1 and TMIE auxiliary subunits (Fig. 1c) to density features of the TMC-1 complex map, in accord with the mass spectrometry data. The CALM-1 subunits 'grip' the cytosolic faces of each TMC-1

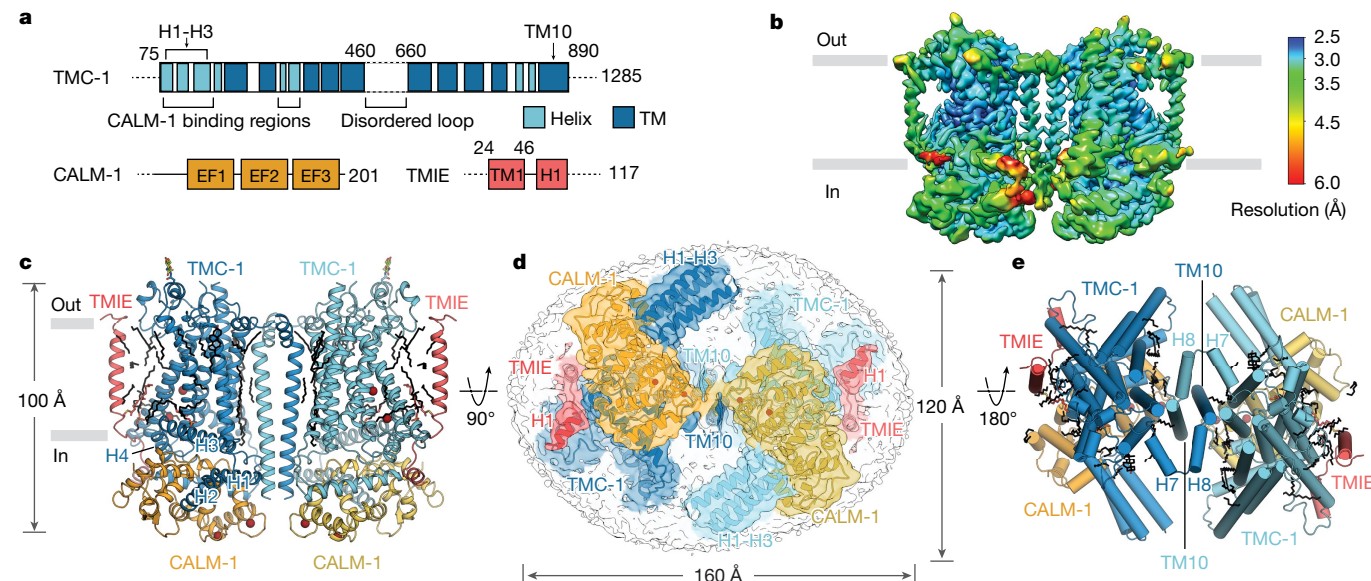

**Fig. 1 | Architecture and subunit arrangement of the TMC-1 complex.**
**a**, Schematic representation of protein constructs that co-purified with TMC-1. EF1–EF3, EF-hand domains; TM, transmembrane domain. **b**, Local resolution map of the native TMC-1 complex after 3D reconstruction. **c**, Overall architecture of the native TMC-1 complex, viewed parallel to the membrane. TMC-1 (dark blue and light blue), CALM-1 (orange and yellow) and TMIE (red and pink) are shown in a cartoon diagram. Lipid-like molecules, *N*-glycans and putative ions are coloured black, green, and dark red, respectively. **d**, Cytosolic view of the reconstructed map fit to the model. Subunit densities are coloured as in **c** and the detergent micelle is shown in grey. **e**, A top-down extracellular view of the TMC-1 complex shows the domain-swapped dimeric interface. α-helices are represented as cylinders.

protomer, and each of the two TMIE subunits span the membrane, nearly 'floating' on the periphery of complex, flanking each TMC-1 subunit. CALM-1 makes extensive contacts with five of the six cytosolic helices, forming a 'cap' at the base of the TMC-1 transmembrane domain. By contrast, the TMIE subunits define the distal edges of the complex, participating in only a handful of protein–protein contacts on the extracellular and cytosolic boundaries of the membrane spanning regions, but with lipid-mediated interactions through the transmembrane regions. Viewed parallel to the membrane and perpendicular to the long face of the complex, the arrangement of subunits resembles an accordion, with the TMIE transmembrane helices forming the instrument handles and the TMC-1 transmembrane domain defining the bellows (Fig. 1c).

## Lipid-mediated interactions of TMIE with TMC-1

TMIE is an essential subunit of the vertebrate MT complex that is necessary for TMC-1 mediated mechanosensory transduction in cochlear hair cells[32] and in zebrafish sensory hair cells[10]. Multiple point mutations in TMIE are linked to deafness (Extended Data Fig. 7), and recent studies suggest a role for TMIE in TMC-1 and TMC-2 localization and channel gating[9–11,33–35]. The *C. elegans* TMC-1 complex contains two copies of TMIE located on the 'outside' of each TMC-1 protomer (Fig. 2a). TMIE consists of a single transmembrane domain followed by an 'elbow-like' linker and a cytosolic helix (Fig. 2b). The flexible, positively charged C-terminal tail was not visible in the cryo-EM map. The interaction between TMIE and TMC-1 is mediated primarily by the cytosolic TMIE elbow, with the highly conserved R49 and R52 forming hydrogen bonds with backbone carbonyl atoms in TMC-1 TM6 and TM8, respectively (Fig. 2c,e). These arginine residues can be mapped to known deafness mutations in humans (R81C and R84W), highlighting the importance of these hydrogen bonds in the TMIE–TMC-1 interaction. Hydrophobic contacts between nonpolar residues in the TMIE elbow and TMC-1 TM6 probably strengthen the complex. Additionally, W25 of TMIE, near the extracellular boundary, contacts L228 in the loop between TMC-1 TM1 and TM2 (Fig. 2d). Mutation of the corresponding residue

in humans (W57) to a stop codon is a cause of deafness[36]. We did not observe density for the N-terminal 17 residues of TMIE in the cryo-EM map, and peptides from this region were not detected in the mass spectrometry analysis, suggesting that the N terminus contains a cleaved signal peptide (Supplementary Fig. 2). N-terminal sequencing of recombinantly expressed mouse TMIE is also consistent with cleavage of a signal peptide (Supplementary Fig. 2), as are truncation experiments of zebrafish TMIE[10], supporting the hypothesis that in *C. elegans* the first approximately 17 residues of TMIE function as a signal peptide.

TMIE and TMC-1 form an intramembraneous cavity that is occupied by multiple detergent or lipid molecules. Several lipids make hydrophobic contacts with nonpolar residues in TMIE and the putative pore-forming TMC-1 helices TM6 and TM8, bridging the two subunits. Consistent with the observed lipid density in the cryo-EM maps, molecular dynamics simulations independently identify multiple lipids in this cavity (Extended Data Fig. 10). Notably, C44 of TMIE on the cytosolic boundary of the transmembrane domain is palmitoylated, with the acyl chain extending along TMC-1 TM8 (Fig. 2d,e). The location of TMIE near the putative TMC-1 pore and its lipid interactions suggests roles for TMIE, and possibly lipids, in gating by sensing membrane tension. This idea is supported by recent studies in mouse cochlear hair cells, which demonstrated that TMIE binds to phospholipids and that its association with lipids is important for TMC-1 mechanosensory transduction[11].

## CALM-1 cloaks cytoplasmic surfaces of TMC-1

CIB2 and its homologue, CIB3, modulate the activity of the MT complex and bind to the TMC-1 subunit[12,14]. Consistent with the role of CIB2 and CIB3 in MT channel function, mutants of CIB2 are associated with non-syndromic hearing loss[25,37,38]. Our mass spectrometry results (Extended Data Fig. 2) demonstrate that CALM-1, the *C. elegans* orthologue of CIB2, co-purifies with TMC-1, consistent with CALM-1 residing within the TMC-1 complex. Inspection of the map of the TMC-1 complex reveals density features for two CALM-1 subunits on the cytosolic faces of each TMC-1 protomer. Using the crystal structure of CIB3

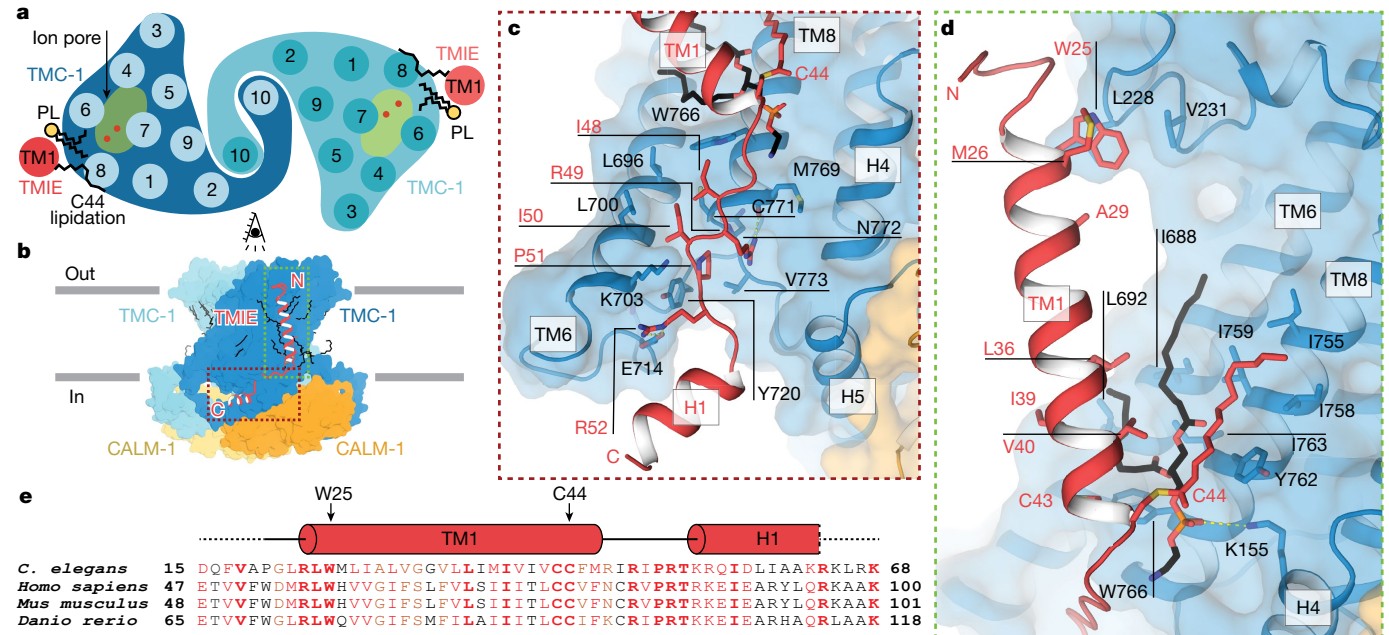

**Fig. 2 | TMIE resides on the periphery of the TMC-1 complex. a**, Schematic representation of TMC-1 (blue) and TMIE (red) transmembrane helices highlights the proximity of TMIE to the putative TMC-1 ion-conduction pathway. Palmitoylation of TMIE C44 and phospholipids (PL) is shown in black. **b**, Overview of the interaction interface between TMIE and TMC-1, viewed from the side. **c**, The interface between the TMIE 'elbow' and TMC-1. Interacting residues are shown as sticks. **d**, The interface between TMIE transmembrane helix and TMC-1, highlighting key residues and lipids. Palmitoylation is shown in red and phospholipid is shown in black. **e**, Multiple sequence alignment of TMIE orthologues. Elements of secondary structure are shown above the sequences and key residues are indicated with black arrows. Residues in black are not conserved, those in red are conservatively substituted, and those in bold red are conserved.

in complex with a TMC-1 peptide[14], we fit models of CALM-1 to their respective density features (Fig. 3a). Similar to other CIB proteins, CALM-1 has three EF-hand motifs, two of which are located proximal to the C terminus and harbour clearly bound $Ca^{2+}$ ions. Following superposition, the root-mean-square deviation between CALM-1 and CIB3 from the CIB3–TMC-1 peptide complex is 0.69 Å, and together with a substantial sequence similarity, underscore the conservation of sequence and structure between the worm and mouse proteins (Supplementary Fig. 3).

Extensive interactions bind CALM-1 and TMC-1 together, involving a buried surface area of around 2,903 Å² and suggesting that CALM-1 may bind to TMC-1 with high affinity (Fig. 3a). Three distinct regions of CALM-1 interact with cytosolic helical features of TMC-1, the first of which involves TMC-1 helices H1 to H3, oriented like 'paddles' nearly parallel to the membrane (Fig. 3b,c). Prominent interactions include side chains in the loop between H1 and H2, which form hydrophobic contacts with CALM-1, together with acidic residues on CALM-1 that create a negatively charged surface juxtaposed to a complementary positively charged surface on the H1–H3 paddle (Fig. 3c). The second binding interface is through a hydrophobic pocket of CALM-1, comprised of its EF-hand motifs, and the cytosolic H5–H6 helices of TMC-1 (Fig. 3d), reminiscent of the CIB3–TMC-1 peptide structure. Aliphatic and aromatic residues, including L308, F309 and Y314 of TMC-1 are docked into the conserved hydrophobic core in CALM-1, further stabilizing the complex by burial of substantial nonpolar surface area. Finally, amino acids D192, R195 and R200 at the C terminus of CALM-1 interact with R780, D313 and E160 of TMC-1, respectively, forming conserved salt bridges through the buried short helix (191–197) of CALM-1 (Fig. 3e). This interface shows that CALM-1 directly engages with the transmembrane helices of TMC-1 via the loop between TM8 and TM9, and is thus positioned to modulate ion channel function.

Multiple missense mutations of human CIB2 or TMC-1 are associated with non-syndromic hearing loss by either impeding the interaction

between TMC-1 and CIB2 or by reducing the $Ca^{2+}$ binding propensity of CIB2[14]. Several of these residues, E178D of human TMC-1 and E64D, F91S, Y115C, I123T and R186W of CIB2, are structurally conserved in the *C. elegans* TMC-1 and CALM-1 complex (Supplementary Fig. 3). Our structure illuminates the proximity of the CALM-1 $Ca^{2+}$ binding sites to the CALM-1 and TMC-1 interface, thus underscoring the roles of both $Ca^{2+}$ and CALM-1 in sculpting the conformation of the TMC-1 and by extrapolation, providing a structural understanding of CIB2 in hair cell function.

Mass spectrometry analysis of the TMC-1 complex indicated the presence of the soluble protein ARRD-6. Upon classification of the single-particle cryo-EM data, we observed a non-two-fold symmetric 3D class defined by an elongated density feature protruding from the CALM-1 auxiliary subunit and we hypothesized that it corresponded to ARRD-6. Arrestins are composed of an N domain and a C domain, each composed of β-sandwich motifs, which together give rise to a protein with an elongated, bean-like shape. We fit the predicted structure of ARRD-6 into the corresponding density feature and although the local resolution of the ARRD-6 region is lower than that of central region of the complex, the fit yielded overall correlation coefficients of 0.69 (mask) and 0.65 (volume) (Extended Data Fig. 5). Moreover, density features for the ARRD-6 β-sheets are clearly observed at the binding interface with CALM-1, as well as for the crossed elongated loops of the N and C domains at the central crest, further supporting the assignment of the density feature to ARRD-6. We observed a 'C-edge loop' structure, positioned at the distal edge of the β-strands in the C domain, a feature which functions as a membrane anchor and is necessary for activation of arrestin[39] (Fig. 3f,g). The C-edge loop of ARRD-6 includes W197 and multiple cysteine residues (Fig. 3g), the latter of which may be palmitoylated and thus poised for membrane anchoring[40]. Additional contacts between CALM-1 and ARRD-6 involve a loop of CALM-1 (P51–K67) with the β-strands in the C domain of ARRD-6 (Fig. 3h). Structural alignment shows that the TMC-1 conformation

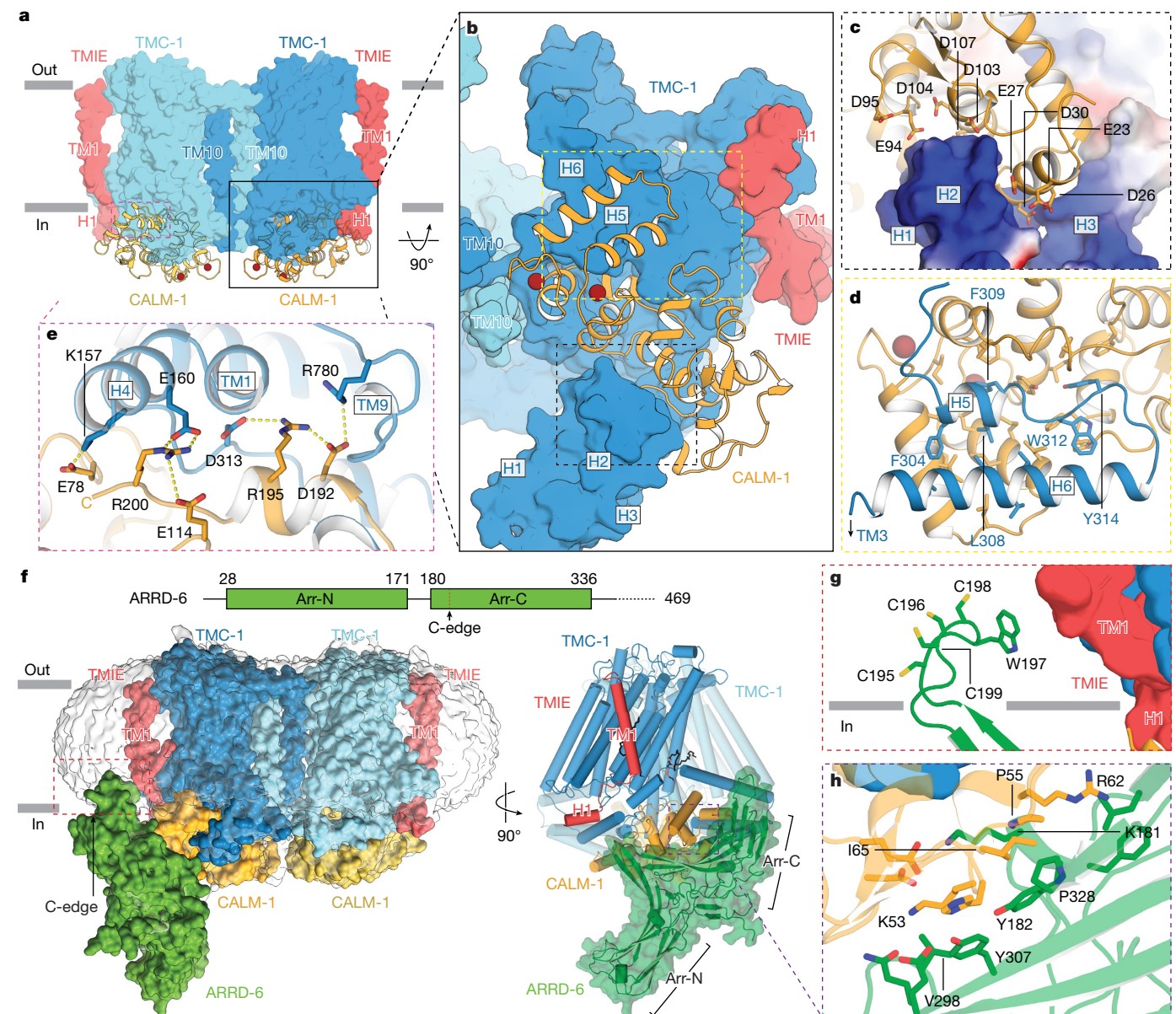

**Fig. 3 | CALM-1 and ARRD-6 auxiliary subunits cap the cytoplasmic face of the TMC-1 complex. a,b**, The binding interface between CALM-1 and TMC-1 viewed parallel to the membrane (**a**) and perpendicular to the membrane (**b**). **c**, Binding interface between CALM-1 and TMC-1 H1–H3. The electrostatic surface of TMC-1 is shown, where blue represents positive regions and red represents negative regions. CALM-1 is shown in yellow. **d**, The interface between CALM-1 and TMC-1 H5 and H6. **e**, Salt bridges between the C terminus of CALM-1 and TMC-1. Putative hydrogen bonds are shown as dashed lines. **f**, Three-dimensional reconstruction of the TMC-1 complex with ARRD-6 viewed

parallel to the membrane. TMC-1, CALM-1, TMIE, and ARRD-6 are shown in blue, yellow, red, and green, respectively. The red dashed rectangle indicates the putative insertion site of the ARRD-6 C-edge loop into the micelle. A schematic diagram of ARRD-6 is shown above the reconstruction, with arrestin N- (Arr-N) and C- (Arr-C) domains. **g**, The interface between the C-edge loop of ARRD-6 and the membrane. ARRD-6 residues that likely participate in membrane interactions are shown as sticks. **h**, The interface between ARRD-6 (green) and CALM-1 (yellow), highlighting residues that are important for the binding interaction.

within the ARRD-6 complex is most similar to the E conformation. The roles of arrestin in the function of the TMC channel of *C. elegans* and in the vertebrate TMC-1 complex remain unknown. We speculate that ARRD-6 may have a regulatory role in TMC-1 channel function or be involved in endocytosis of the TMC-1 complex by recruitment of cytoskeleton proteins, similar to the role of α-arrestin in the regulation of G-protein-coupled receptors[41]. At this juncture, we do not know why we observed only a single ARRD-6 subunit bound to the complex, as there is sufficient space for two. One subunit may be unbound from the complex or the second subunit may be only partially occupied. Further experiments are required to address these questions.

## Mapping the putative ion channel pore

Single-channel currents measured from cochlear hair cells demonstrate that the mammalian MT complex is cation-selective[42] with a high permeability for $Ca^{2+}$. TMC-1 or TMC-2, are the likely pore-forming subunits of the mammalian MT complex, and cysteine mutagenesis experiments have identified several pore-lining residues that are critical for TMC-1-mediated mechanosensory transduction[6,7] (Extended Data Fig. 7a). *C. elegans* TMC-1 mediates mechanosensitivity in worm OLQ neurons and body wall muscles[13], but its ion selectivity and permeation properties are not known, largely owing to challenges associated with

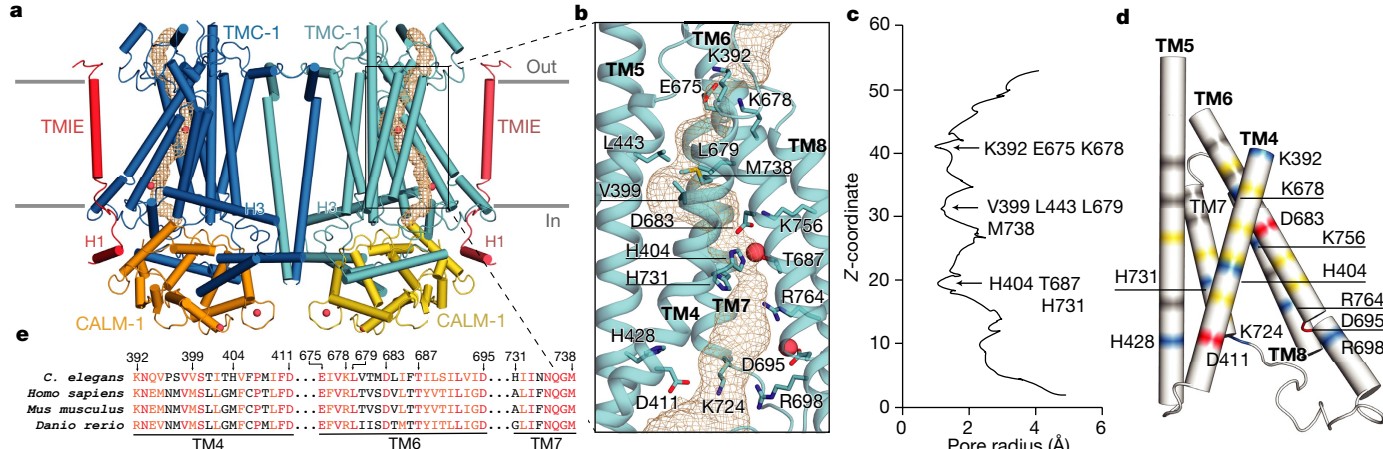

**Fig. 4 | The putative ion-conduction pore of TMC-1. a**, The location of the pore (gold mesh) is shown in the context of the TMC-1 complex. Putative calcium ions are shown as red spheres. **b**, An expanded view of the ion permeation pathway, highlighting pore-lining residues, shown as sticks, and putative ions (red). **c**, van der Waals radius of the pore plotted against the distance along the pore axis, calculated by MOLE 2.0. **d**, The electrostatic potential of pore-lining residues is depicted in different colours: grey, nonpolar; yellow, polar; red, acidic; and blue, basic. Acidic and basic residues are labelled. **e**, Multiple sequence alignment of selected residues from TMC-1 putative pore-forming helices.

heterologous expression of the recombinant complex and with vanishingly small amounts of native material. Notably, *C. elegans* and mouse TMC-1 also function as Na⁺-permeable leak channels that modulate the resting membrane potential via a depolarizing background leak conductance, suggesting that TMC-1 may serve multiple cellular roles[23,43] and indicating that the channel pore is permeable to a greater diversity of ions than previously appreciated.

To gain insight into the nature and function of the *C. elegans* TMC-1 ion-conduction pathway, we superimposed the TMC-1 subunit onto the structures of TMEM16A and OSCA1.2, revealing a similar architecture among the transmembrane domains (Extended Data Fig. 6). The TMEM16A and OSCA1.2 dimer assemblies contain two pores—one within each subunit—that are defined by helices TM3–TM7. Structural similarities between TMEM16a and OSCA1.2 suggest that TMC-1 may also have two pores and could conduct ions through a structurally analogous pathway composed of TM4–TM8 (Fig. 4a). The putative ion-conduction pathway appears to be closed, with a narrow pore blocked by three constrictions (Fig. 4b,c). Polar and basic residues line the first constriction site near the extracellular pore entrance and nonpolar residues dominate the second constriction site, located approximately 20 Å further down the conduction pathway, towards the cytoplasm. The location of the second constriction site aligns well with the narrow 'neck' region of the putative OSCA1.2 pore, which is composed primarily of hydrophobic residues, as well as with the hydrophobic TMEM16a gate[28,44]. The remaining 40 Å of the conduction pathway is lined mostly by polar and charged residues (Fig. 4d). Seven basic residues line the pore, two of which (H404 and H731) partially define the third and narrowest constriction site. Cysteine mutagenesis experiments have identified eight pore-lining residues in mouse TMC-1[7,45] and, although five of these residues are not conserved in *C. elegans*, the locations of all eight map to pore-lining positions in *C. elegans* TMC-1 (Extended Data Fig. 7a), thus supporting their location in the ion-conduction pathway. We visualized two spherical, non-protein densities near two acidic residues (D683 and D695) that may correspond to bound cations (Fig. 4b). At the present resolution of the structure, however, we cannot determine whether these features are Ca²⁺ ions. Both asparagine residues are conserved in human TMC-1, suggesting that they are important for ion coordination. Although the ionizable residues lining the closed pore are predominately basic and thus not in keeping with a canonical Ca²⁺-permeable channel, which typically harbours acidic residues, it remains unknown which residues line the permeation pathway in the open conformation. Additionally, the overall residue composition is similar to that of the mechanosensitive ion channel OSCA1.2[28]. OSCA1.2 displays stretch-activated non-selective cation currents with 17–21% Cl⁻ permeability[46], suggesting that *C. elegans* TMC-1 may exhibit similar permeation properties.

To visualize the ion-conduction pore of the vertebrate MT complex, we exploited the structure of the *C. elegans* complex and constructed a homology model of the human TMC-1 complex that includes TMC-1, CIB2 and TMIE (Extended Data Fig. 8). Upon inspection of this structure, we found that the putative pore is lined by two basic residues and five acidic residues, in keeping with the channel being permeable to Ca²⁺. In addition, there are relatively more polar residues compared to the worm orthologue and the histidine residues that occlude the second constriction site in *C. elegans* TMC-1 are replaced by M418 and A579 in the human model. The vertebrate MT complex also endows hair cells with permeability to organic molecules, including the dye FM1–43[47]. Although our structure does not provide direct insight into the pathway of small molecule permeation, several hydrophobic crevices—including the lipid-lined space between TMC-1 and TMIE—provide possible routes for the transmembrane passage of small molecules such as FM1–43.

## Expanded and contracted conformations

We observed a second conformation of the TMC-1 complex by 3D classification, designated as the C conformation (Extended Data Fig. 9). The TMC-1 subunits in the E and C conformations have a similar overall structure, both including closed ion channels. In the C conformation, however, the TM10 helix is bent approximately 9° compared with that in the E conformation, and there is one additional helical turn of TM10 in the E conformation. Upon successively superimposing the TMC-1 subunits from the C and E complexes, we observed that in the E state, each half of the TMC-1 complex, composed of TMC-1, CALM-1 and TMIE subunits, is rotated by around 8° compared with the C state, by way of an axis of rotation that is located near the TMC-1 H7 and H8 helices and oriented approximately parallel to the membrane. The movement of each half of the complex—when viewed parallel to the membrane plane—thus resembles the motion of an accordion, with the cytoplasmic regions of the complex undergoing relatively larger conformational displacements compared with those on the extracellular side of the membrane. Indeed, in comparing the C and E states, the amphipathic TMC-1 H3 helices move farther apart by approximately 11 Å, thus underscoring the magnitude of the conformational change. As described below, these changes are accompanied by significant alterations in

the structure and mechanical behaviour of the membrane surrounding the protein (Extended Data Fig. 9). Further studies, however, will be necessary to determine their ramifications on the function of the MT complex. Nevertheless, these results illustrate the conformational plasticity of the TMC-1 complex and, reciprocally, the possibility that deformations of the membrane may induce conformational changes in the TMC-1 complex.

## Membrane embedding of the TMC-1 complex

To understand how the TMC-1 complex interacts with individual lipids as well as with the lipid bilayer, we performed all-atom and coarse-grained molecular dynamics simulations on both the E and C complexes embedded in membranes composed of phospholipids and cholesterol (Extended Data Fig. 10a). The all-atom set included three independent simulations for each conformational state of the protein, yielding a collective sampling time of 6 µs, whereas the coarse-grained runs were performed on larger membrane patches each including four copies of TMC-1 in either E or C conformations (Extended Data Fig. 10a), and each simulated for 10 µs resulting in a collective sampling time of 80 µs of lipid–protein interactions. Examination of the equilibrated membrane structure around the TMC-1 complex indicates a deep penetration and anchoring of the amphipathic, paddle H3 helix into the cytosolic leaflet of the bilayer (Fig. 5a,b). In agreement with the cryo-EM density maps, the simulations show that phospholipids and cholesterol occupy the cavity between TMIE and TMC-1, and cholesterol is enriched in crevices near the two-fold-symmetry related, TM10 helices at the TMC-1 subunit interface, together supporting the importance of lipids in the structure and function of the complex (Extended Data Fig. 10b,c). The TMC-1 complex also distorts the membrane bilayer, promoting both thinning and thickening of the membrane in its vicinity, with especially prominent thinning of the cytoplasmic leaflet within the region of H3 helix insertion (Fig. 5c and Extended Data Fig. 10d). Of note, the E and C conformations generate distinct membrane deformation patterns (Fig. 5c and Extended Data Figs. 9b and 10d,e). In particular, the E conformation induces a more pronounced membrane distortion (thickening and thinning) close to the protein, resulting in a longer-range membrane deformation propagation (around 50 Å away from the protein) (Extended Data Fig. 10d,e).

## Summary

The molecular structures of the TMC-1 complex reveal the identity, architecture and membrane association of key subunits central to vertebrate and *C. elegans* mechanosensory transduction. The accordion-shaped, two-fold symmetric complex contains TMIE subunits poised like handles perpendicular to the membrane, and amphipathic TMC-1 H3 helices inserted and parallel to the membrane plane, each providing possible mechanisms for direct or indirect transduction of force to ion channel gating, respectively. Interactions between TMIE and TMC-1 are restricted to the extracellular and intracellular boundaries of TMIE TM1, resulting in a large intramembranous cavity between the two subunits that we hypothesize is filled with lipid molecules in a plasma membrane environment. TMIE is positioned proximal to the putative pore-forming helices TM4–8 (Fig. 2a), interacting directly with TMC-1 TM6 and TM8 through hydrogen bonds and via the palmitoyl group attached to TMIE residue C44. The arrangement of subunits resembles an accordion, with TMIE forming the instrument handles. We speculate that force applied to TMIE, either via the membrane or by way of direct contacts with an auxiliary subunit, is then coupled to the TMC-1 pore-forming transmembrane helices, opening the pore to allow ion permeation.

In vertebrates, protocadherin-15 transduces force to stereocilia tips, opening the MT channel. Previous studies have suggested that protocadherin-15 forms a stable, dimeric complex with LHFPL5 yet

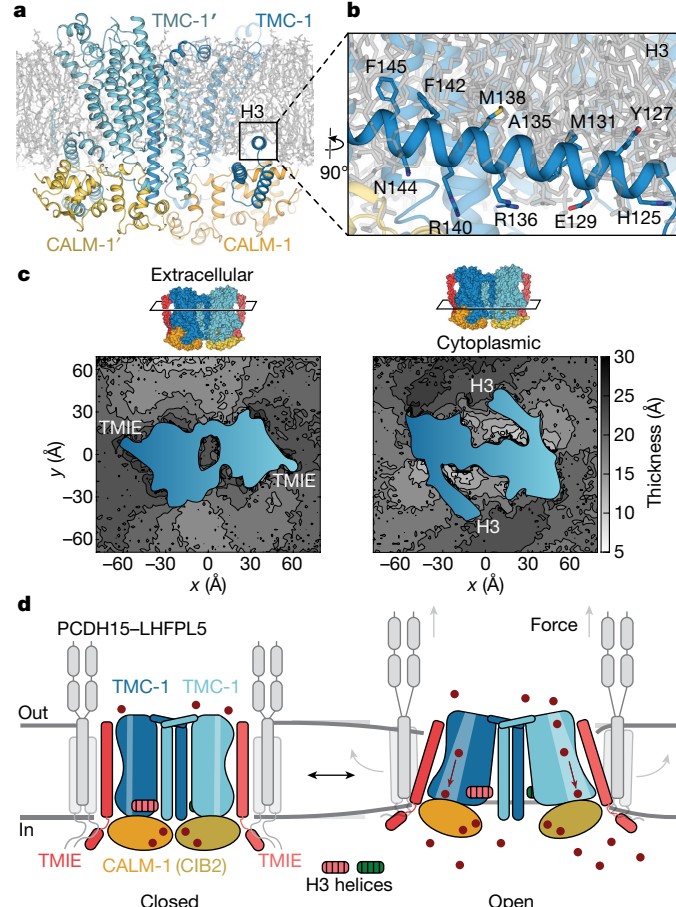

**Fig. 5 | Membrane integration and mechanism. a**, Molecular dynamics simulation of the membrane-embedded TMC-1 complex (E conformation) shows penetration of the H3 helix into the lipid bilayer. **b**, Key residues that define the amphipathic nature of the H3 helix are shown as sticks. **c**, Thickness patterns for extracellular and cytosolic membrane leaflets averaged over the last 500 ns of three simulated replicas for the E conformation. The pattern for the C conformation is shown in Extended Data Fig. 8. The cross-section of the protein is shown in blue and the location of the cross-section is indicated above the plots using a surface representation of the TMC-1 complex. **d**, Schematic illustrating mechanisms by which direct or indirect forces might be transduced to ion channel gating. Grey arrows (right) show how membrane tension could directly gate the TMC-1 complex by exerting force on TMIE. Indirect force as a result of changes in membrane thickness could affect the position of the membrane-embedded helix H3, modulating ion channel gating.

also interacts with TMC-1 and TMIE subunits[9,15,17,48,49], but how this interaction might occur remains unknown. One possibility is that the protocaderin-15 dimer is situated coincident with the two-fold axis of the TMC-1 complex, with procadherin-15 transmembrane helices surrounding the TMC-1 TM10 helixes. This closed symmetric dimeric complex would enable tension on protocadherin-15 to be directly transduced to the TMC-1 complex via the protocadherin-15 contacts with the TM10 helices. Alternatively, protocadherin-15 dimers could interact with TMIE helices, with one protocadherin subunit interacting with a single TMIE subunit, thus forming an open complex in which the unpaired protocadherin-15 subunit could interact with a TMIE subunit from another TMC-1 complex (Fig. 5d). This model both provides a direct mechanism for force transduction from protocadherin-15 to TMIE and then to the TMC-1 ion channel pore and provides a mechanism for the clustering of TMC-1 complexes[50]. In addition to the direct transduction of force, we also speculate that H3 of the TMC-1 subunit acts like a paddle in the membrane that moves up or down as the membrane

thins or thickens, thus providing a mechanism for force coupling to the channel via the membrane. Further studies of open-channel conformations of the *C. elegans* TMC-1 complex, in addition to structures of the vertebrate MT complex, will be required to more fully elucidate the mechanisms of force transduction. Nevertheless, these TMC-1 complexes provide a framework for structure-based mechanisms of touch in *C. elegans* and of hearing and balance in vertebrates.

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

## Methods

### Transgenic worm design

The strain PHX2173 *tmc-1(syb2173)* was generated by SunyBiotech using CRISPR–Cas9 genome editing and is referred to as the *tmc-1::mVenus* line (Supplementary Fig. 1). The TMC-1-mVenus-3×Flag sequence was inserted prior to the stop codon of the endogenous *tmc-1* gene (Wormbase: T13G4.3.1). The genotype was confirmed using PCR and primers ER02-seq-s (ATTAGATCCCGCAAGAGAAT) and ER02-seq-a (AAGGTGA-TATGAACGAACCG), which bind 452 bp upstream and 408 bp downstream from the insertion site, respectively, to amplify the region of interest. The PCR product was subsequently sequenced using primers ER02-mid-s (CATGAAGCAACACGACTTCT) and ER02-mid-a (TCTTC-GATGTTGTGACGGAT), which bind within the TMC-1-mVenus-3×Flag sequence. To enable elution of the engineered TMC-1 complex from affinity chromatography resin, a PreScission protease (3C) cleavage site was placed between the C terminus of TMC-1 and the mVenus fluorophore.

### Spectral confocal imaging

Adult worms were immobilized in M9 buffer (22 mM $KH_2PO_4$, 42 mM $Na_2HPO_4$, 86 mM NaCl and 1 mM $MgCl_2$) containing 30 mM sodium azide and placed on slides that were prepared with ~4 mm agar pads. Spectral images were acquired on a Zeiss 34-channel LSM 880 Fast Airyscan inverted microscope with a 40× 1.2 NA water-immersion objective lens. Linear unmixing was employed to distinguish between the mVenus signal and autofluorescence. The autofluorescence signal was subtracted from each image. The 3D *z*-stack information is presented in 2D after performing a maximum intensity projection.

### Large scale *C. elegans* culture

All *C. elegans* strains were maintained and grown according to Wormbook methods (http://www.wormbook.org). For large scale liquid culture, nematode growth medium (NGM) agar plates were prepared and spread with *E. coli* strain HB101, allowing the bacterial lawn to grow overnight at 37 °C. Worms were transferred to the NGM plates and grown for 3–4 days at 20 °C until HB101 cells were depleted. Worms on the plates were transferred to a liquid medium in 2 l baffled flasks, supplemented with HB101 (~15 g per 500 ml medium) and streptomycin (50 µg ml⁻¹), and worms were grown at 20 °C with vigorous shaking (150 rpm) for 70–72 h. To collect worms, the liquid culture flasks were placed on ice for 1 h to allow the worms to settle. The media was removed, and the worm slurry was collected in a tube, washed twice with 50 ml of ice cold M9 buffer by successive centrifugation (800*g* for 1 min) and resuspension. Worms were 'cleaned' by sucrose density centrifugation at 1,500*g* for 5 min after bringing the volume of worm slurry up to 25 ml with M9 buffer and adding 25 ml of ice cold 60% (w/v) sucrose. The worm layer on top was recovered and placed in a new tube and then washed twice with 50 ml of ice cold M9 buffer. The volume of the worm pellet was measured and the same volume of M9 buffer was added to the tube and worm balls were made by dripping the slurry into liquid nitrogen. The worm balls were stored at −80 °C until further use.

### Isolation of the native TMC-1 complex

Approximately 80 g of frozen worm balls were disrupted using a ball mill (MM400, Retch) where the grinding jar and ball were pre-cooled in liquid nitrogen. Disrupted worm powder was solubilized at 4 °C for 2 h in a buffer containing 50 mM Tris-Cl (pH 9.3), 50 mM NaCl, 5 mM EDTA, 2% (w/v) glyco-diosgenin (GDN), and protease inhibitors (0.8 µM aprotinin, 2 µg ml⁻¹ leupeptin and 2 µM pepstatin). After centrifugation at 40,000 rpm (186,000*g*) for 50 min, the supernatant was applied to anti-Flag M2 affinity resin (A2220, Sigma-Aldrich) and incubated overnight on a rotator at 4 °C. The resin was washed 5 times with a buffer containing 20 mM Tris-Cl (pH 8.5), 150 mM NaCl and 0.02% (w/v) GDN, using a volume of buffer that was 200-fold the volume of the resin. The TMC-1 complex was eluted by incubating with 40 µg of 3C protease at 4 °C for 4 h on the rotator. Subsequently, the solution was supplemented with 3 mM $CaCl_2$, final concentration, and the eluate was filtered with a 0.22 µm centrifuge tube filter. The concentrate was loaded onto a SEC column (Superose 6 Increase 10/30 GL, GE Healthcare), equilibrated in a buffer composed of 20 mM Tris-Cl (pH 8.5), 150 mM NaCl, 0.02% (w/v) GDN and 3 mM $CaCl_2$. The peak fractions from the putative dimeric TMC-1 complex were pooled and concentrated for cryo-EM grid preparation. Approximately 50 ng of TMC-1 was isolated from 80 g of worm balls, which translates to approximately $6 \times 10^7$ worms. The amount of protein was determined via mVenus fluorescence based on a standard plot. The estimated total amount of the TMC-1 complex including TMC-1, CALM-1 and TMIE is 60 ng. The isolated native TMC-1 sample was analysed by SDS–PAGE and the protein bands were visualized by silver staining. For mass spectrometry analysis, the putative dimeric TMC-1 complex peak was pooled and concentrated to a volume of 50 µl for further use. The same isolation method was used to make the wild-type worm sample from the *C. elegans* N2 strain for use as a control in the mass spectrometry experiments in order to evaluate non-specific binding of *C. elegans* proteins to anti-Flag M2 affinity resin.

### Isolation of the native TMC-1 complex for SiMPull

The native TMC-1 complex, bound to anti-Flag M2 affinity resin, was eluted with a buffer comprising 20 mM Tris-Cl (pH 8.5), 150 mM NaCl and 0.02% (w/v) GDN, supplemented with 1 mg ml⁻¹ 2×Flag peptide at 4 °C for 40 min on a rotator. The eluate was concentrated and subjected to further purification on a SEC column. The putative dimeric TMC-1 complex peak was pooled and used for SiMPull.

### SiMPull

Coverslips and glass slides were cleaned, passivated and coated with a solution consisting of 50 mM methoxy polyethylene glycol (mPEG) and 1.25 mM biotinylated PEG in water. A flow chamber was created by drilling 0.75 mm holes in a quartz slide and by placing double-sided tape between the holes. A coverslip was placed on top of the slide and the edges were sealed with epoxy, creating small flow chambers. A solution of phosphate buffered saline (PBS) that included 0.25 mg ml⁻¹ streptavidin was then applied to the slide, allowed to incubate for 5 min, and washed off with a buffer consisting of 50 mM Tris, 50 mM NaCl and 0.25 mg ml⁻¹ bovine serum albumin (BSA), pH 8.0 (T50 BSA buffer). Biotinylated anti-GFP nanobody in T50 BSA at 10 µg ml⁻¹ was applied to the slide, allowed to incubate for 10 min, and washed off with 30 µl buffer A (20 mM Tris, pH 8.0, 150 mM NaCl, 0.02% (w/v) GDN, 3 mM $CaCl_2$).

The TMC-1 complex was isolated as described in 'Isolation of the native TMC-1 complex for SiMPull'. The complex was purified by SEC, diluted 1:200, and immediately applied to the chamber. After a 5-min incubation, the slide was washed with 30 µl buffer A and the chamber was imaged using a Leica DMi8 TIRF microscope with an oil-immersion 100× objective. Images were captured using a back-illuminated EMCCD camera (Andor iXon Ultra 888) with a 133 × 133 µm imaging area and a 13 µm pixel size. This 13 µm pixel size corresponds to 130 nm on the sample owing to the 100× objective. To estimate non-specific binding to the glass slide, the purified TMC-1 complex was applied to a separate chamber wherein the anti-GFP nanobody was not included and the other steps remained identical. The observed spot count from this chamber was used to estimate the number of background fluorescence spots.

Photobleaching movies were acquired by exposing the imaging area for 60 s. To count the number of TMC-1 subunits, single-molecule fluorescence time traces of the mVenus-tagged TMC-1 complex were generated using a custom python script. Each trace was manually scored as having one to three bleaching steps or was discarded if no clean bleaching steps could be identified. The resulting distribution of bleaching steps closely matches a binomial distribution for a dimeric

protein based on an estimated GFP maturation of 80%. A total of 600 molecules were evaluated from 3 separate movies. Scoring was verified by assessing the intensity of the spot; on average, the molecules that bleach in two steps were twice as bright as those that bleach in one step.

## Mass spectrometry

The purified TMC-1 complex sample was dried, dissolved in 5% sodium dodecyl sulfate, 8 M urea, 100 mM glycine (pH 7.55), reduced with (tris(2-carboxyethyl)phosphine at 37 °C for 15 min, alkylated with methyl methanethiosulfonate for 15 min at room temperature followed by addition of acidified 90% methanol and 100 mM triethylammonium bicarbonate buffer (TEAB; pH 7.55). The sample was then digested in an S-trap micro column briefly with 2 µg of a Tryp/LysC protease mixture, followed by a wash and 2 h digestion at 47 °C with trypsin. The peptides were eluted with 50 mM TEAB and 50% acetonitrile, 0.2% formic acid, pooled and dried. Each sample was dissolved in 20 µl of 5% formic acid and injected into Thermo Fisher QExactive HF mass spectrometer. Protein digests were separated using liquid chromatography with a Dionex RSLC UHPLC system, then delivered to a QExactive HF (Thermo Fisher) using electrospray ionization with a Nano Flex Ion Spray Source (Thermo Fisher) fitted with a 20um stainless steel nano-bore emitter spray tip and 1.0 kV source voltage. Xcalibur version 4.0 was used to control the system. Samples were applied at 10 µl min⁻¹ to a Symmetry C18 trap cartridge (Waters) for 10 min, then switched onto a 75 µm x 250 mm NanoAcquity BEH 130 C18 column with 1.7 µm particles (Waters) using mobile phases water (A) and acetonitrile (B) containing 0.1% formic acid, 7.5–30% acetonitrile gradient over 60 min and 300 nl min⁻¹ flow rate. Survey mass spectra were acquired over $m/z$ 375–1400 at 120,000 resolution ($m/z$ 200) and data-dependent acquisition selected the top 10 most abundant precursor ions for tandem mass spectrometry by higher energy collisional dissociation using an isolation width of 1.2 $m/z$, normalized collision energy of 30 and a resolution of 30,000. Dynamic exclusion was set to auto, charge state for MS/MS +2 to +7, maximum ion time 100 ms, minimum AGC target of $3 \times 10^6$ in MS1 mode and $5 \times 10^3$ in MS2 mode. Data analysis was performed using Comet (v. 2016.01, rev. 3)[51] against a January 2022 version of canonical FASTA protein database containing *C. elegans* UniProt sequences and concatenated sequence-reversed entries to estimate error thresholds and 179 common contaminant sequences and their reversed forms. Comet searches for all samples performed with trypsin enzyme specificity with monoisotopic parent ion mass tolerance set to 1.25 Da and monoisotopic fragment ion mass tolerance set at 1.0005 Da. A static modification of +45.9877 Da was added to all cysteine residues and a variable modification of +15.9949 Da on methionine residues. A linear discriminant transformation was used to improve the identification sensitivity from the Comet analysis[52,53]. Separate histograms were created for matches to forward sequences and for matches to reversed sequences for all peptides of seven amino acids or longer. The score histograms of reversed matches were used to estimate peptide false discovery rates (FDR) and set score thresholds for each peptide class. The overall protein FDR was 1.2%.

## Cryo-EM sample preparation

A volume of 3.5 µl of the concentrated TMC-1 complex was applied to a Quantifoil grid (R2/1 300 gold mesh, covered by 2 nm continuous carbon film), which was glow discharged at 15 mA for 30 s in the presence of amylamine. The grids were blotted and flash frozen using a Vitrobot mark IV for 2.5 s with 0 blot force after 30 s wait time under 100% humidity at 15 °C. The grids were plunge-frozen into liquid ethane, cooled by liquid nitrogen.

## Data acquisition

The native TMC-1 complex dataset was collected on a 300 keV FEI Titan Krios microscope equipped with a K3 detector. The micrographs were acquired in super-resolution mode (0.4195 Å per pixel) with a magnification of 105,000× corresponding to a physical pixel size of 0.839 Å per pixel. Images were collected by a 3×3 multi-hole per stage shift and a 6 multi-shot per hole method using Serial EM, with a defocus range of −1.0 to −2.4 µm. Each movie stack was exposed for 3.3 s and consisted of 50 frames per movie, with a total dose of 50 e⁻ Å⁻². A total of 26,055 movies were collected.

## Image processing

Beam-induced motion was corrected by patch motion correction with an output Fourier cropping factor of 1/2 (0.839 Å per pixel). Contrast transfer function (CTF) parameters were estimated by patch CTF estimation in CryoSparc v3.3.1[54]. A total of 25,852 movies were selected by manual curation and the particles were picked by using blob-picker with minimum and maximum particle diameters of 140 Å and 200 Å, respectively. Initially, 7.9 million particles were picked and extracted with a box size of 400 pixels and binned 4× (3.356 Å per pixel). After one round of 2D classification, 'junk' particles were removed, resulting in 3.2 million particles in total. The particles with the highest resolution features, approximately 1.5 million, were used for ab initio reconstruction. The full particle stack consisting of 3.2 million particles from 2D classification were then subjected to heterogeneous refinement using the reconstructed models from the ab initio reconstruction. Probable monomeric TMC-1 complexes, detergent micelles and additional junk particles were removed in this step, yielding 1.65 million particles. Particles were then re-extracted from unbinned images. Subsequently, heterogeneous refinement using C1 symmetry was performed with the re-extracted 1.65 million particles, yielding 8 classes. Among them, three good classes composed of 667,000 particles were selected and used for further analysis. After one round of heterogeneous refinement with 4 classes in C2 symmetry, two classes containing 208,000 and 199,000 particles were discerned, each with distinct features and that we describe as the contracted and expanded forms, respectively. One more round of heterogeneous refinement was performed for both particle stacks to sort out groups of homogeneous particles from each class. To attain higher resolution and improved map quality, non-uniform refinement including defocus and global CTF refinement was performed in Cryosparc v3.3.1 of each individual class, with particle stack sizes of 141,000 (contracted) and 142,000 (expanded), resulting in resolutions at 3.09 Å and 3.10 Å, respectively.

Among the initial 8 classes from the heterogeneous refinement of 1.65 million particles, one of the classes, which contained 272,000 particles, had an additional density feature, proximal to CALM-1. Further heterogeneous refinement and 3D classification without alignment was carried out with this class to sort out heterogeneous particles. One more round of heterogeneous refinement in Cryosparc resulted in one promising particle class, containing 99k particles, out of four total classes. Non-uniform refinement, including defocus and global CTF refinement, was performed with the selected class, resulting in a map at 3.54 Å resolution. To improve the density of unknown protein bound to CALM-1, local refinement in Cryosparc was performed using a mask, covering the 'extra density' and CALM-1.

## Structure determination and model building

The initial electron microscopy density map was sharpened with Phenix AutoSharpen[55], and both sharpened and unsharpened maps were used for structure determination. Various strategies including de novo building, structure prediction, docking and homologous modelling were used for model building. The transmembrane helices of TMC-1 (TM1–TM9, excluding TM10), predicted by Alphafold2[56] as a template, were fit into the map with rigid body fitting in UCSF Chimera[57] and de novo model building using Coot[58]. The possible ion permeation pore of the channel was determined by MOLE 2.0[59]. Carbohydrate groups were modelled to protruding densities of N209 on TMC-1, at a predicted *N*-linked glycosylation site.

To build the structure of CALM-1 into the expanded conformation density map of the TMC-1 complex, we exploited the previously determined structure of CIB3 in complex with a TMC-1 peptide (Protein Data Bank ID: 6WUD). We docked CIB3 into the density map using rigid body fitting in UCSF Chimera, using the highly conserved H5 and H6 helices of TMC-1 as a guidepost, and proceeded by introducing the sequence of CALM-1 into the model, followed by manual adjustment of the model using Coot. Conserved bulky side chains, including F84, Y129, and F197, that protrude into hydrophobic cavities and are facing the helices of TMC-1, facilitated the definition of the correct register of the CALM-1 sequence.

The auxiliary subunit, TMIE, was built manually into the density map of the expanded conformation using Coot. The bulky side chain density of tryptophan (W25) and lipid modification on cysteine residue (C44) helped to assign the sequence register in the context of the density map. The model was refined against the sharpened map by real-space refinement in Phenix.

The following regions of TMC-1 were not modelled into the map because of weak or absent densities: The N-terminal region of TMC-1 (M1 to P73), the predicted loop region between TM5 and TM6 (S460 to N663) and the C-terminal region (L886 to D1285). The side chains with weak density on H1 (75–87) and TM10 (870–885) helices were modelled as alanine residues. The N-terminal region of CALM-1, from residues 1 to 17, and the amino acids of TMIE, including 1–17 and 64–117, were not modelled due to a lack of density. As discussed in the main text, we suggest that residues 1–17 of TMIE comprise a signal peptide.

For the modelling of the unknown density on CALM-1 we speculated that ARRD-6 was a possible candidate auxiliary protein based on the mass spectrometry results. Although the overall map quality of the putative ARRD-6 region was not sufficient for de novo model building, we could find several β-sheets with side chain densities on the map. Using the predicted structure of ARRD-6 and the crossed-protrusion of two loops of the N- and C- domains of arrestin (82–85 of the N, and 249–256 of the C domain), we could align the predicted ARRD-6 model into the unknown density, thus providing further evidence that the unknown density is ARRD-6. The estimated local resolution of ARRD-6 density ranges between 4–7 Å and the calculated Q-score of ARRD-6 model-to-map from MapQ[60] plugin in Chimera is 0.25, which corresponds to the estimated resolution of 4.91 Å, suggesting that the model is reasonably placed in the map. The final CC of the ARRD-6 and overall model are 0.42 and 0.69, respectively. All figures for density maps and models were generated by Pymol and ChimeraX.

## Molecular dynamics simulations

Molecular dynamics simulations were performed on both the C and E conformations and at two different resolutions, coarse-grained (CG) and all-atom (AA). Starting from the cryo-EM modelled structure, a C-terminal carboxylic cap group, an N-terminal ammonium capping group, missing side chains, and all the hydrogen atoms were modelled using the PSFGEN plugin of VMD[61]. PROPKA was employed to estimate the pKa of titratable residues[62,63]. All titratable side chains were found in their default protonation state. The modelled structures were then used for setting up the CG and AA simulations.

## Coarse-grained simulation setup

The Martini-based CG models[64–66] of the TMC-1 complexes were generated, employing the Martinize protocol as described in the Martini website (http://www.cgmartini.nl/), followed by applying an elastic network on atom pairs within a 10 Å cut-off. The CG parameters for the palmitoylated Cys in TMIE was obtained from a previous work[67]. The initial orientation of the protein in the membrane was adopted from the orientations of proteins in membranes (OPM) database. The protein complexes were then inserted in a lipid bilayer composed of palmitoyl-oleoyl-phosphatidyl-ethanolamine (PE), palmitoyl-oleoyl-phosphatidyl-choline (PC), sphingomyelin (SM),

and cholesterol with a molar ratio of 54:32:8:6. The sphingomyelin used in the CG simulations was the DPSM lipid in Martini lingo, corresponding to C(d18:1/18:0) N-stearoyl-D-erythro tails. In the AA simulations, sphingomyelin was PSM in CHARMM forcefield corresponding to N-palmitoyl-sphingomyelin (PSM). The secondary structures of the proteins were derived from the AA models and maintained throughout the CG simulations. To enhance the sampling and improve statistics, four copies of the protein in each conformation were embedded in a large patch ($400 \times 400$ Å$^2$) of lipid bilayer at an inter-protein distance of 200 Å, using the computer program Insane[68]. The systems were then solvated and ionized with 150 mM NaCl employing Insane (system size: 330,000 CG beads).

## All-atom simulation setup

The CG-equilibrated protein–membrane complexes after 8 μs of CG simulations were back-mapped to CHARMM-based AA models, employing CHARMM-GUI[69,70], followed by isolating one of the four replicas (a protein copy with membrane padding of approximately 40 Å) from the larger membrane patch. DOWSER was used to internally hydrate the protein[71,72]. The protein–membrane systems were then solvated with water including 150 mM NaCl in VMD (system size: 340,000 atoms). To improve the statistics and further reduce any bias from the initial lipid placement, three independent membrane systems, with independently placed initial lipids, were generated for each conformation using the Membrane Mixer plugin (MMP)[73].

## Coarse-grained simulation protocol

CG systems were simulated using GROMACS[74], with the standard Martini v2.2 simulation parameters[66]. The simulations were conducted with a 20 fs timestep. The temperature was fixed at 310 K using velocity-rescaling thermostat[75] with a time constant of 1 ps for coupling. A semi-isotropic, 1 bar pressure was maintained by the Berendsen barostat[76] with a compressibility of $3 \times 10^{-4}$ bar and a relaxation time constant of 5 ps. The systems were initially energy minimized for 1,000 steps, followed by relaxation runs of 18 ns, while the lipid bilayer headgroups and protein backbones were restrained harmonically. During the initial 18 ns, the restraints applied to bilayer headgroups were removed stepwise (from $k = 200$ kJ mol$^{-1}$ nm$^{-2}$ to zero), while the restraints on the protein backbone ($k = 1,000$ kJ mol$^{-1}$ nm$^{-2}$) were unchanged. The 4-protein systems in each conformation were then simulated for 10 μs, with restraints only applied to the protein backbones, resulting in a cumulative sampling of 80 μs (4 copies × 2 conformations × 10 μs).

## All-atom simulation protocol

The AA converted systems were simulated using the following protocol: (1) 5,000 steps of minimization, followed by 5 ns of relaxation, during which the proteins' heavy atoms as well as the bound Ca$^{2+}$ ions were harmonically restrained ($k = 10$ kcal mol$^{-1}$ Å$^{-2}$) to their position in the cryo-EM model; (2) 1 ns of equilibration with harmonic restraints only on the protein backbone heavy atoms ($k = 10$ kcal mol$^{-1}$ Å$^{-2}$). The coordination of Ca$^{2+}$ ions in this step was maintained by the application of the Extra Bonds algorithm in NAMD[77,78], (3) 200 ps of equilibration during which the restraints on the backbone were maintained whereas the Extra Bonds on the Ca$^{2+}$ ions were removed; (4) two additional replicas were generated with the MMP plugin and 1 μs of production runs were performed on each of the three independent simulation replicas while only the protein backbone heavy atoms were restrained. Steps 1–3 were performed using NAMD2[77,78]. The 1-μs production runs for all three replicas were conducted on Anton2[79].

All AA simulations were performed using the fully atomistic CHARMM36m[80] and CHARMM36[81] force fields for the proteins and lipids, respectively. Water molecules were modelled with TIP3P[82]. In NAMD simulations, a 12 Å cut-off was used for short-range, non-bonded interactions, with switching distance starting at 10 Å. Particle mesh Ewald (PME) was used to calculate long-range electrostatic

interactions[83] with a grid density of 1 Å$^{-1}$, and a PME interpolation order of 6. The SHAKE algorithm was used to constrain bonds involving hydrogen atoms[84]. Temperature was kept constant at 310 K using Langevin thermostat with a damping coefficient of 1.0 ps$^{-1}$. Pressure was maintained at 1 atm employing the Nosé–Hoover Langevin piston barostat with period and decay of 100 and 50 fs, respectively[85,86]. All systems were simulated in a flexible cell allowing the dimensions of the periodic cell to change independently while keeping the aspect ratio in the $xy$ plane (membrane plane) constant. The timestep was set to 2 fs, and the PME and Lennard–Jones forces were updated at every other and each timestep, respectively.

For Anton2 simulations, 310 K temperature and 1 bar pressure were kept by the Nosé–Hoover chain coupling and Martyna–Tuckerman–Klein schemes[85], as implemented using a multigrator scheme[87]. M-SHAKE was used to constrain all the bonds to hydrogen atoms[88], and a 2.5 fs timestep was used in all the simulations. The long-range electrostatic interactions were calculated by employing the fast Fourier transform (FFT) method on Anton2[79].

### Membrane thickness and lipid distribution analysis

The effect of the protein on the thickness of each membrane leaflet was quantified in both CG and AA simulations by monitoring the $z$ (membrane normal) distance of the phosphate groups of phospholipids with respect to the bilayer midplane, over the second half of each trajectory (last 5 μs of the CG simulations or the last 500 ns of the AA simulations). The thickness values were plotted using a histogram with $2 \times 2$ Å$^2$ bins in the $xy$ plane (membrane plane), for each leaflet individually. Cholesterol and phospholipid distributions were similarly calculated by summing the positions of the hydroxy (for cholesterol) or phosphate (for phospholipids) beads over the last 5 μs of the CG trajectories into histograms.

### Lipid depletion–enrichment analysis

First, individual lipid counts for all lipid species within 7 Å (using cholesterol hydroxyl or phospholipid phosphate beads) of the four protein copies over the 10 μs of the CG simulation were determined. A depletion–enrichment index for lipid type L was then defined using the following equation[89]:

$$\text{depletion–enrichment index (L)} = \frac{\text{ratio(L)}_{7A}}{\text{ratio(L)}_{bulk}},$$

where ratio(L)$_{7A}$ is the number of molecules of lipid type L within 7 Å of protein copies as a fraction of the total number of lipid molecules within 7 Å of protein copies, and ratio(L)$_{bulk}$ is the number of molecules of lipid type L in the membrane as a fraction of the total number of lipid molecules in the membrane.

### Homology modelling of human TMC-1 complex

The cryo-EM structure of the E conformation of *C. elegans* TMC-1 complex (containing six chains: two TMC-1, two CALM-1 and two TMIE) was used as a template to build a homology model of human TMC-1 complex. Each chain in the template structure was isolated and its sequence was aligned to the corresponding human sequence with AlignMe[90]. The aligned sequences were then used in the multi-chain capability of MODELLER[91] to generate a human TMC-1 complex. The discrete optimized protein energy (DOPE)[92] and GA341[93,94] methods were used to assess the quality of the generated model. The optimization was performed with a maximum iteration of 300 and the model with the best molecular probability density function (molpdf) was selected (Extended Data Fig. 7). The entire optimization cycle was repeated twice to obtain a better structure.

### Reporting summary

Further information on research design is available in the Nature Research Reporting Summary linked to this article.

## Data availability

The coordinates and volumes for the cryo-EM data have been deposited in the Electron Microscopy Data Bank under accession codes EMD-26741 (expanded conformation), EMD-26742 (contracted conformation) and EMD-26743 (with ARRD-6). The coordinates have been deposited in the Protein Data Bank under accession codes 7USW (expanded conformation), 7USX (contracted conformation), and 7USY (with ARRD-6). All the initial and final snapshots of the molecular dynamics trajectories, as well as simulation parameters and configuration files are deposited at https://doi.org/10.5281/zenodo.6780283.

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

**Acknowledgements** We thank T. Nicolson, P. Barr-Gillespie, D. Farrens, M. Mayer, A. Aballay, J. Ge, J. Elferich and members of the Gouaux and Baconguis laboratories for helpful discussions; S. Petrie and B. Jenkins for help with worm spectral imaging; T. Provitola for assistance with figures; A. Reddy for mass spectrometric analysis; J. Meyers and S. Yang for help with cryo-EM screening and data collection; A. Chinn for help with worm growth; and R. Hallford for proof reading. Initial cryo-EM grids were screened at the Pacific Northwest Cryo-EM Center (PNCC), which is supported by NIH grant U24GM129547 and performed at the PNCC at OHSU, accessed through EMSL (grid.436923.9), a DOE Office of Science User Facility sponsored by the Office of Biological and Environmental Research. The large single-particle cryo-EM dataset was collected at the Janelia Research Campus of the Howard Hughes Medical Institute (HHMI). The OHSU Proteomics Shared Resource is partially supported by NIH core grants P30EY010572 and P30CA069533. This work was supported by NIH grant 1F32DC017894 to S.C. The simulations were supported by the NIH grants, P41-GM104601 and R01-GM123455 to E.T. Simulations were performed using allocations on Anton at Pittsburgh Supercomputing Center (award MCB100017P to E.T.), and XSEDE resources provided by the National Science Foundation Supercomputing Centers (XSEDE grant number MCA06N060 to E.T.). E.G. gratefully acknowledges J. LaCroute and B. LaCroute for support, and is an investigator of the HHMI.

**Author contributions** H.J., S.C. and A.G. performed the experiments. H.J., S.C. and A.G., together with E.G., designed the project and wrote the manuscript. S.D.-G., A.R. and E.T. performed and analysed molecular dynamics simulations. All authors contributed to manuscript preparation.

**Competing interests** The authors declare no competing interests.

**Additional information**
**Correspondence and requests for materials** should be addressed to Eric Gouaux.

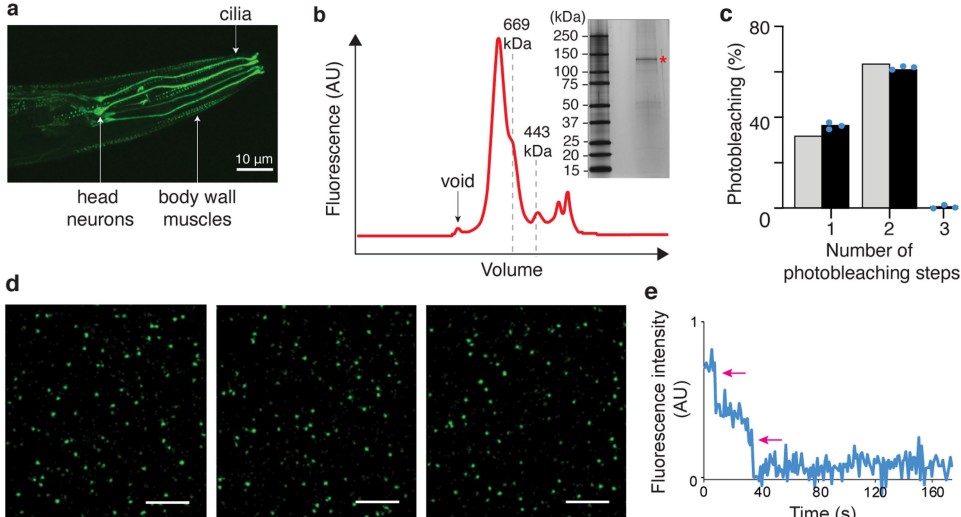

**Extended Data Fig. 1 | Dimeric TMC-1 complex from *C. elegans* copurifies with additional proteins. a**, Spectral confocal image of mVenus fluorescence in an adult *tmc-1::mVenus* worm showing mVenus fluorescence in the head neurons, cilia and body wall muscles. Shown is one representative image of five total images. **b**, Representative FSEC profile of the TMC-1 complex, detected via the mVenus tag. Inset shows a silver-stained, SDS-PAGE gel of the purified TMC-1 complex. Red asterisk indicates TMC-1. The experiments were repeated two times with similar results. Thyroglobulin (669 kDa) and apoferritin (443 kDa) were used for protein molecular mass standards. **c**, The distribution of mVenus photobleaching steps for the TMC-1 complex is consistent with a binomial function (grey bars) an assembly with two fluorophores. A total of n = 600 spots were analysed from three photobleaching movies (200 spots per movie) at random locations in the imaging chamber. Each movie is represented by a blue dot. **d**, Images are shown for the SEC-purified mVenus-tagged TMC-1 complex captured with biotinylated anti-GFP nanobody. Scale bar = 5 μm. **e**, Representative trace showing the two-step photobleaching (red arrows) of the mVenus-tagged TMC-1 complex.

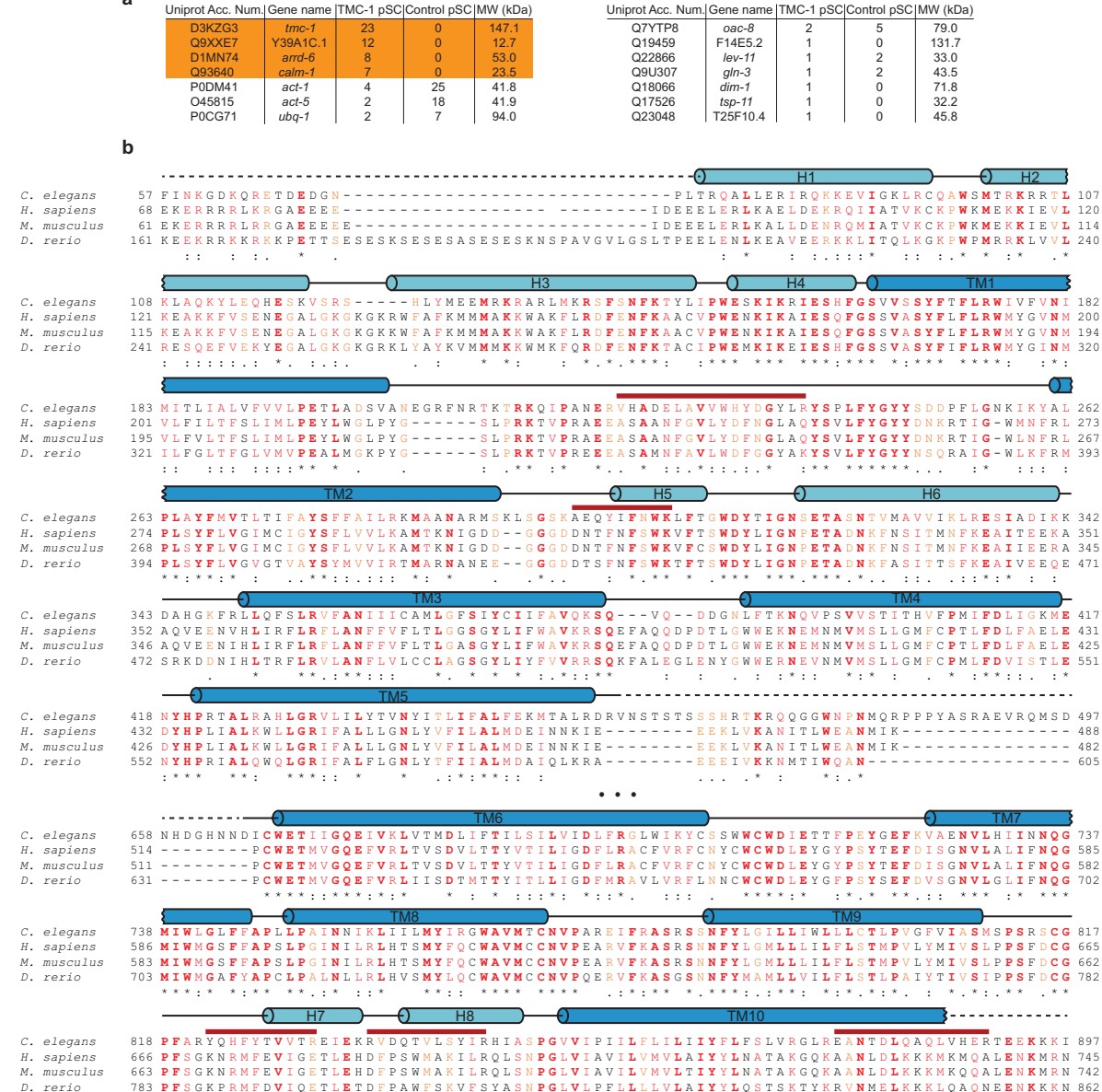

| Uniprot Acc. Num. | Gene name | TMC-1 pSC | Control pSC | MW (kDa) |
|---|---|---|---|---|
| D3KZG3 | tmc-1 | 23 | 0 | 147.1 |
| Q9XXE7 | Y39A1C.1 | 12 | 0 | 12.7 |
| D1MN74 | arrd-6 | 8 | 0 | 53.0 |
| Q93640 | calm-1 | 7 | 0 | 23.5 |
| P0DM41 | act-1 | 4 | 25 | 41.8 |
| O45815 | act-5 | 2 | 18 | 41.9 |
| P0CG71 | ubq-1 | 2 | 7 | 94.0 |

| Uniprot Acc. Num. | Gene name | TMC-1 pSC | Control pSC | MW (kDa) |
|---|---|---|---|---|
| Q7YTP8 | oac-8 | 2 | 5 | 79.0 |
| Q19459 | F14E5.2 | 1 | 0 | 131.7 |
| Q22866 | lev-11 | 1 | 2 | 33.0 |
| Q9U307 | gln-3 | 1 | 2 | 43.5 |
| Q18066 | dim-1 | 1 | 0 | 71.8 |
| Q17526 | tsp-11 | 1 | 0 | 32.2 |
| Q23048 | T25F10.4 | 1 | 0 | 45.8 |

**Extended Data Fig. 2 | MS analysis of the TMC-1 complex. a**, Proteins detected by MS, via their associated peptide fragments, are listed with their gene name and molecular mass. The number of identified unique peptides from both the native TMC-1 complex and from wild-type worms (*C. elegans* N2), used as a control, are also indicated. **b**, Amino acid sequence and secondary structure of *C. elegans* TMC-1 are shown. The secondary structure based on the cryo-EM structure is indicated above the sequences as cylinders (α-helices), black lines (loop regions), or dashed lines (disordered residues). Red lines above the sequences indicate *C. elegans* peptides found by MS. . Note that the TMC-1 segments, corresponding to the sequence of 13–33, 557–566, 567–587, 877–890, 897–904, 917–927, 972–996, 1041–1052, 1177–1190, 1192–1216, and 1261–1269 are also found by MS, but not indicated in **b**.

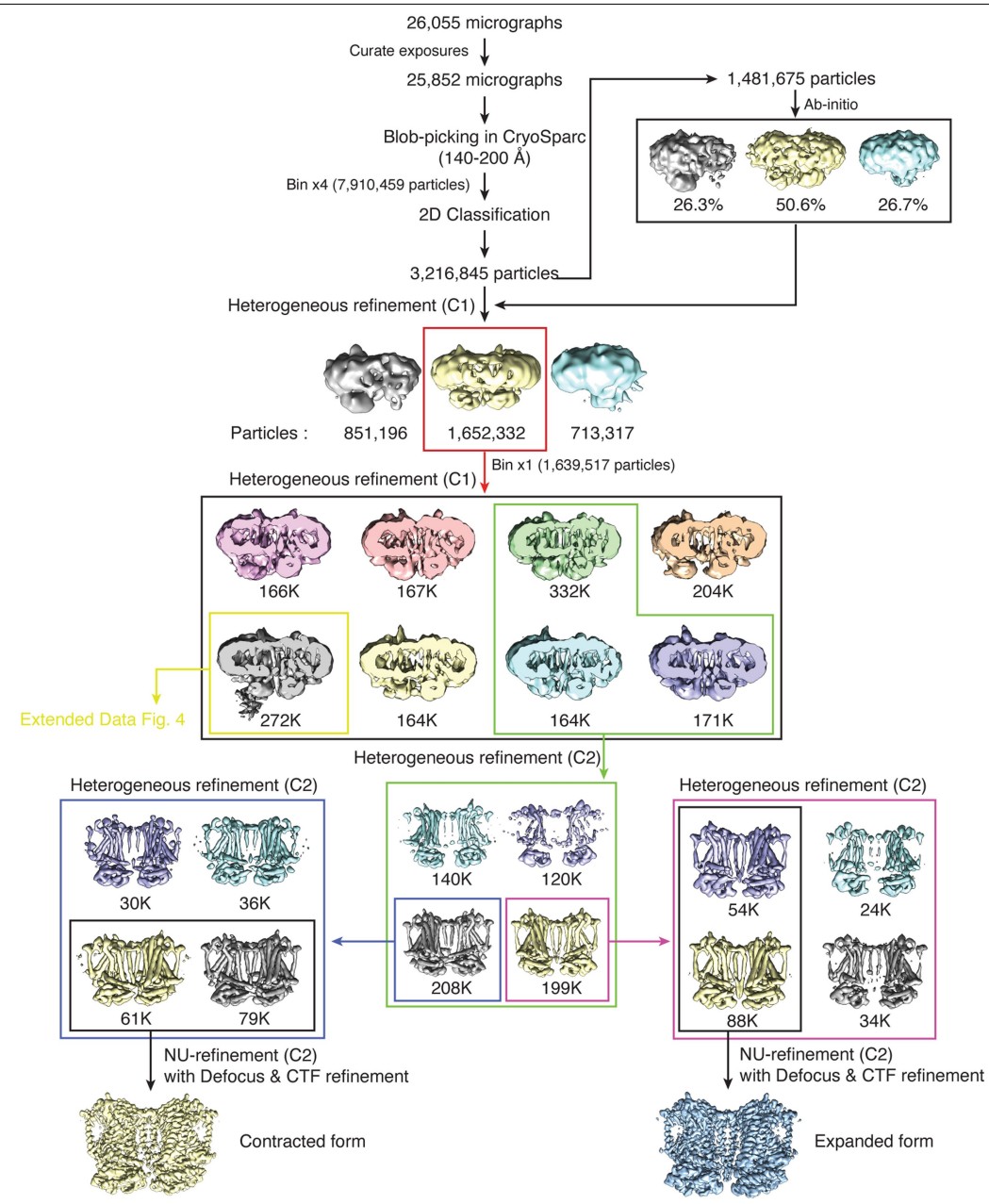

**Extended Data Fig. 3 | Cryo-EM processing workflow of E and C conformations.** Flow chart for cryo-EM data analysis of E and C conformation of the TMC-1 complex.

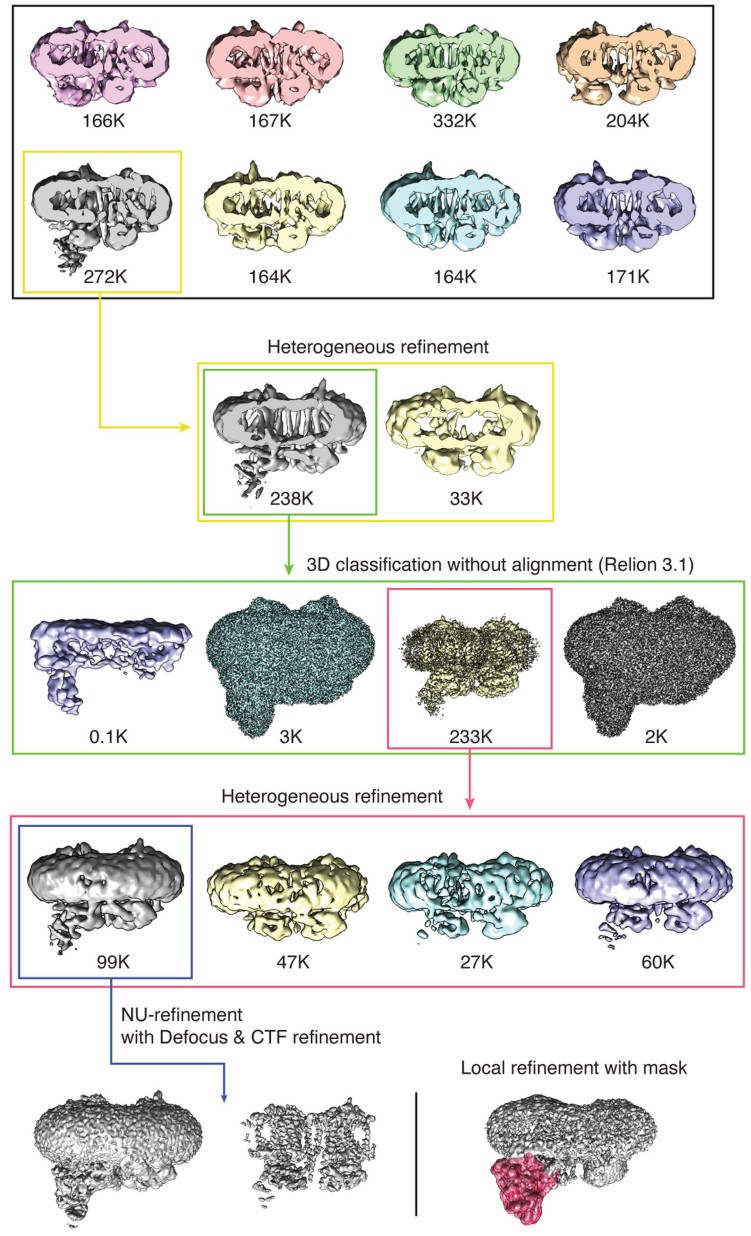

**Extended Data Fig. 4 | Cryo-EM processing workflow of TMC-1 complex with ARRD-6.** Flow chart for cryo-EM data analysis of the TMC-1 complex with ARRD-6.

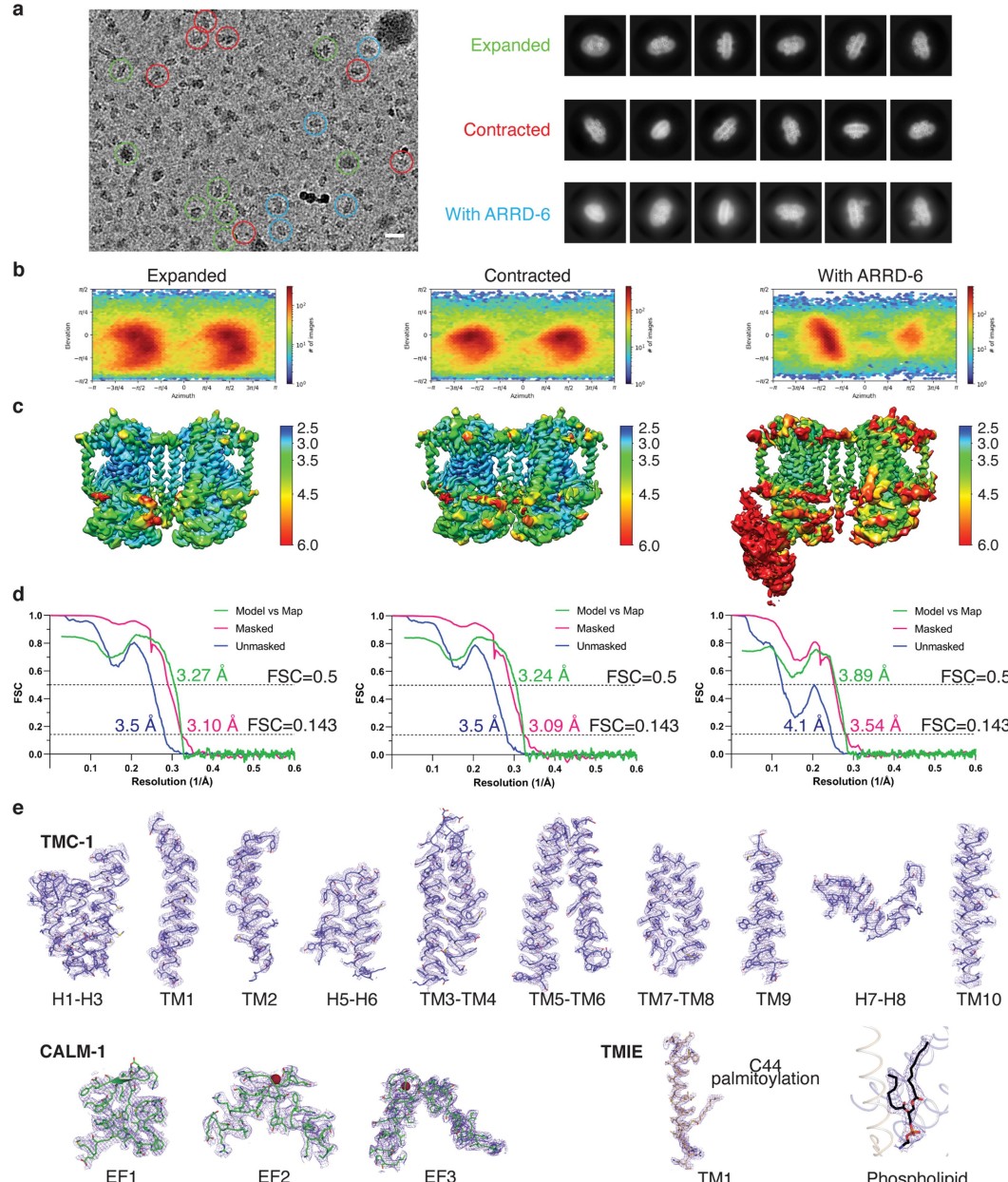

**Extended Data Fig. 5 | Cryo-EM classes, statistics, angular distributions and selected sections of density maps. a**, A representative cryo-EM micrograph of the TMC-1 complex together with several 2D classes of each major 3D class. Each different colored circle (green, red, and blue) on the micrograph indicates particles that were classified to each conformation-Expanded, Contracted, and With ARRD-6, respectively. Scale bar = 200 Å. **b**, Angular distributions of final reconstructions. **c**, Electron density map of each model colored by local resolution values. **d**, Fourier shell correlations (FSC) curve for each model. **e**, Fragments of cryo-EM density map and atomic model of TMC-1 and each auxiliary subunit. The cryo-EM maps are shown as purple mesh.

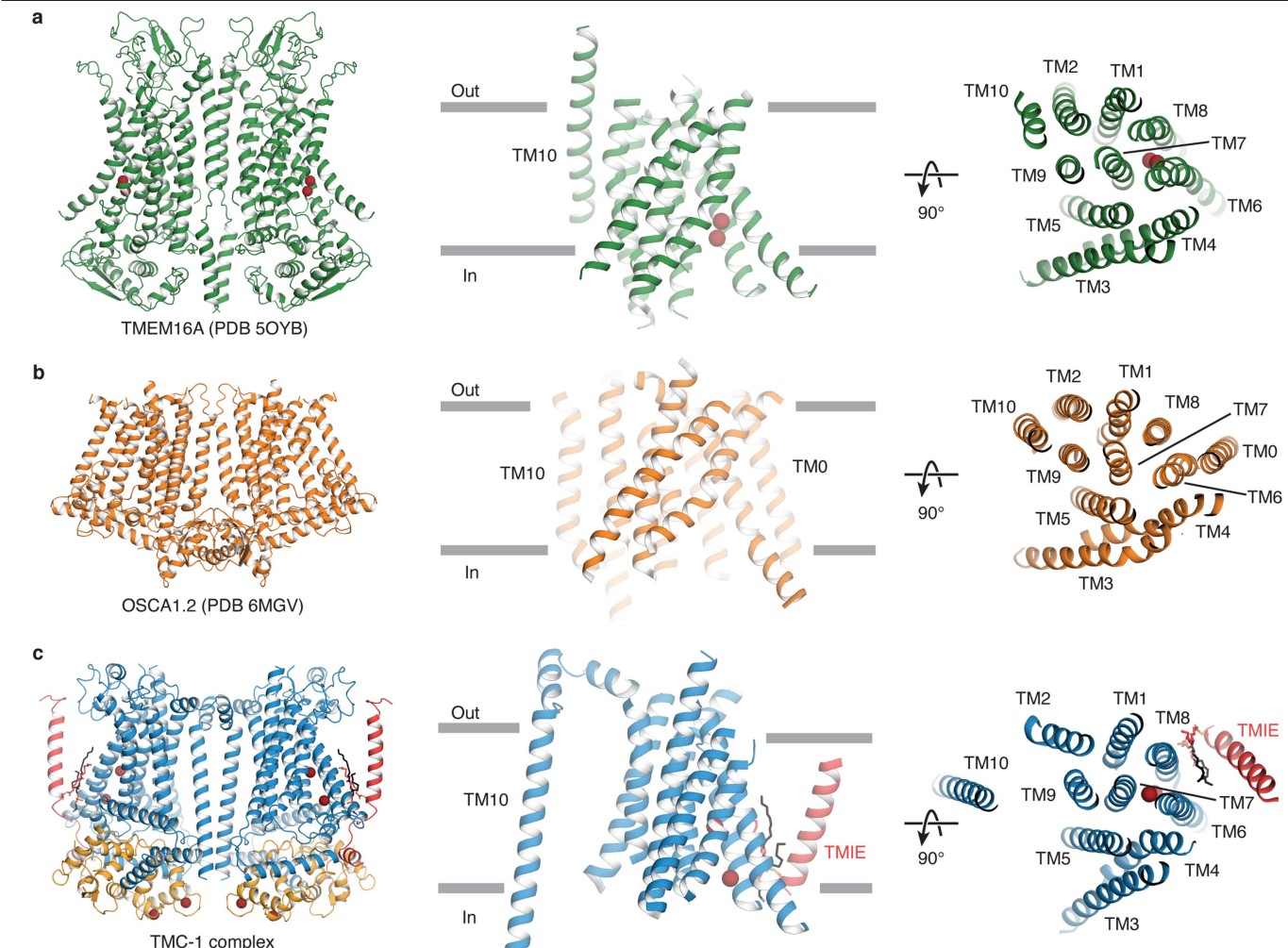

**Extended Data Fig. 6 | Structural comparison of TMEM16, OSCA1.2, and the TMC-1 complex. a**, **b**, **c**. Structures of TMEM16A (5OYB), OSCA1.2 (6MGV), and the TMC-1 complex viewed from the same relative perspectives. The side views of the transmembrane regions and the top-down views are shown in the cartoon model. Putative ions are shown as red spheres.

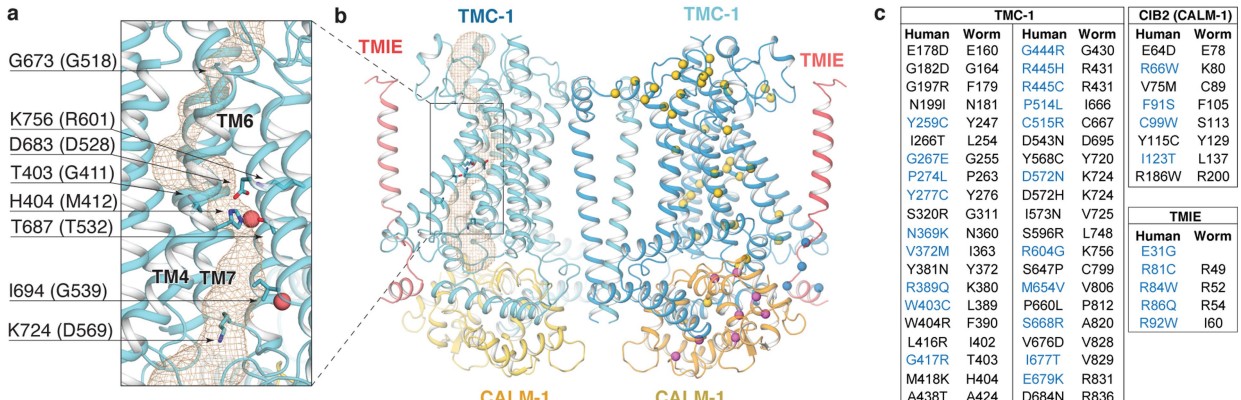

**Extended Data Fig. 7 | The locations of key mutations mapped onto the TMC-1 complex. a**, The locations of cysteine mutations that perturb TMC-1 single channel conductance in mice are shown on the *C. elegans* TMC-1 complex. The *C. elegans* residue number is labeled and the corresponding mouse residue is indicated in parentheses. **b**, The structure of the TMC-1 complex and the locations of mutations associated with hearing loss or deafness. Cα positions of the residues in question are shown as yellow (TMC-1), purple (CALM-1), or blue (TMIE) spheres. **c**, A table of the residues shown in panel b. Mutations linked to hearing loss and deafness in humans are colored black and blue, respectively.

Panel c table:

| TMC-1 | | | | CIB2 (CALM-1) | |
|---|---|---|---|---|---|
| Human | Worm | Human | Worm | Human | Worm |
| E178D | E160 | G444R | G430 | E64D | E78 |
| G182D | G164 | R445H | R431 | R66W | K80 |
| G197R | F179 | R445C | R431 | V75M | C89 |
| N199I | N181 | P514L | I666 | F91S | F105 |
| Y259C | Y247 | C515R | C667 | C99W | S113 |
| I266T | L254 | D543N | D695 | Y115C | Y129 |
| G267E | G255 | Y568C | Y720 | I123T | L137 |
| P274L | P263 | D572N | K724 | R186W | R200 |
| Y277C | Y276 | D572H | K724 | | |
| S320R | G311 | I573N | V725 | **TMIE** | |
| N369K | N360 | S596R | L748 | Human | Worm |
| V372M | I363 | R604G | K756 | E31G | |
| Y381N | Y372 | S647P | C799 | R81C | R49 |
| R389Q | K380 | M654V | V806 | R84W | R52 |
| W403C | L389 | P660L | P812 | R86Q | R54 |
| W404R | F390 | S668R | A820 | R92W | I60 |
| L416R | I402 | V676D | V828 | | |
| G417R | T403 | I677T | V829 | | |
| M418K | H404 | E679K | R831 | | |
| A438T | A424 | D684N | R836 | | |

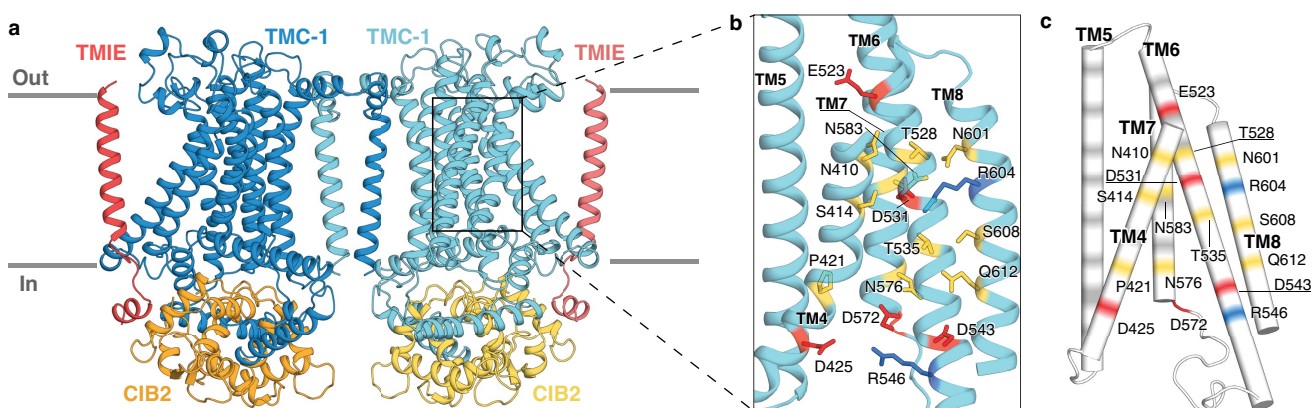

**Extended Data Fig. 8 | Residue composition of the putative ion conduction pathway of the human TMC-1 complex homology model. a**, Homology model of human TMC-1 complex: TMC-1 (dark blue and light blue), CIB2 (orange and yellow), and TMIE (red and pink). **b**, An expanded view of the putative ion conduction pathway, highlighting pore-lining residues. Polar (yellow), acidic (red), and basic (blue) residues are shown as sticks. **c**, Electrostatic potential of pore-lining residues are depicted in different colors: grey = nonpolar, yellow = polar, red = acidic, blue = basic. Acidic, basic, and polar residues are labeled.

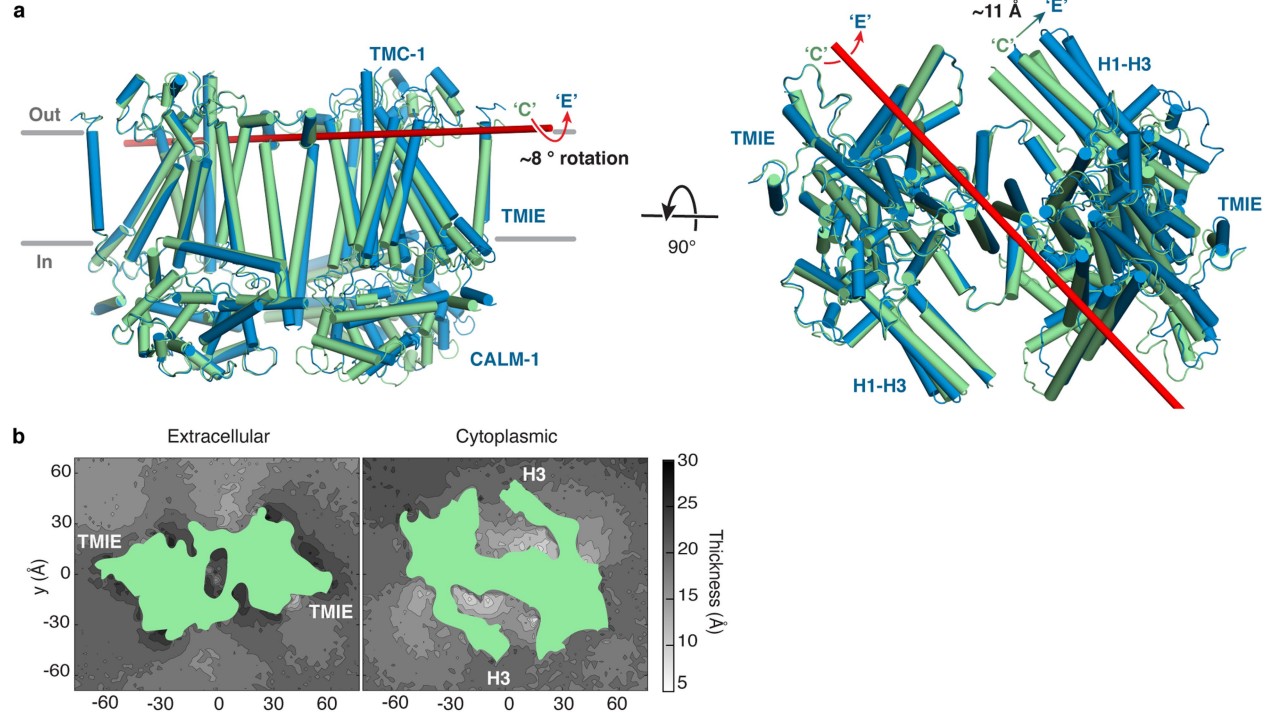

**Extended Data Fig. 9 | Expanded (E) and Contracted (C) conformations of the TMC-1 complex. a**, A TMC-1 protomer in the E conformation (blue) and the C conformation (green), superposed based on backbone alpha-carbon atoms, highlighting conformational changes in TMC-1, as well as CALM-1 and TMIE. The axis of rotation is shown as a red bar and arrows indicate the direction of rotation from the C to the E state. **b**, Membrane thickness in the C conformation calculated from the last 500 ns of the atomistic MD trajectories, averaged over all the three simulation replicas. Heatmaps corresponding to the thickness of extracellular and cytoplasmic leaflets are shown in left and right panels, respectively.

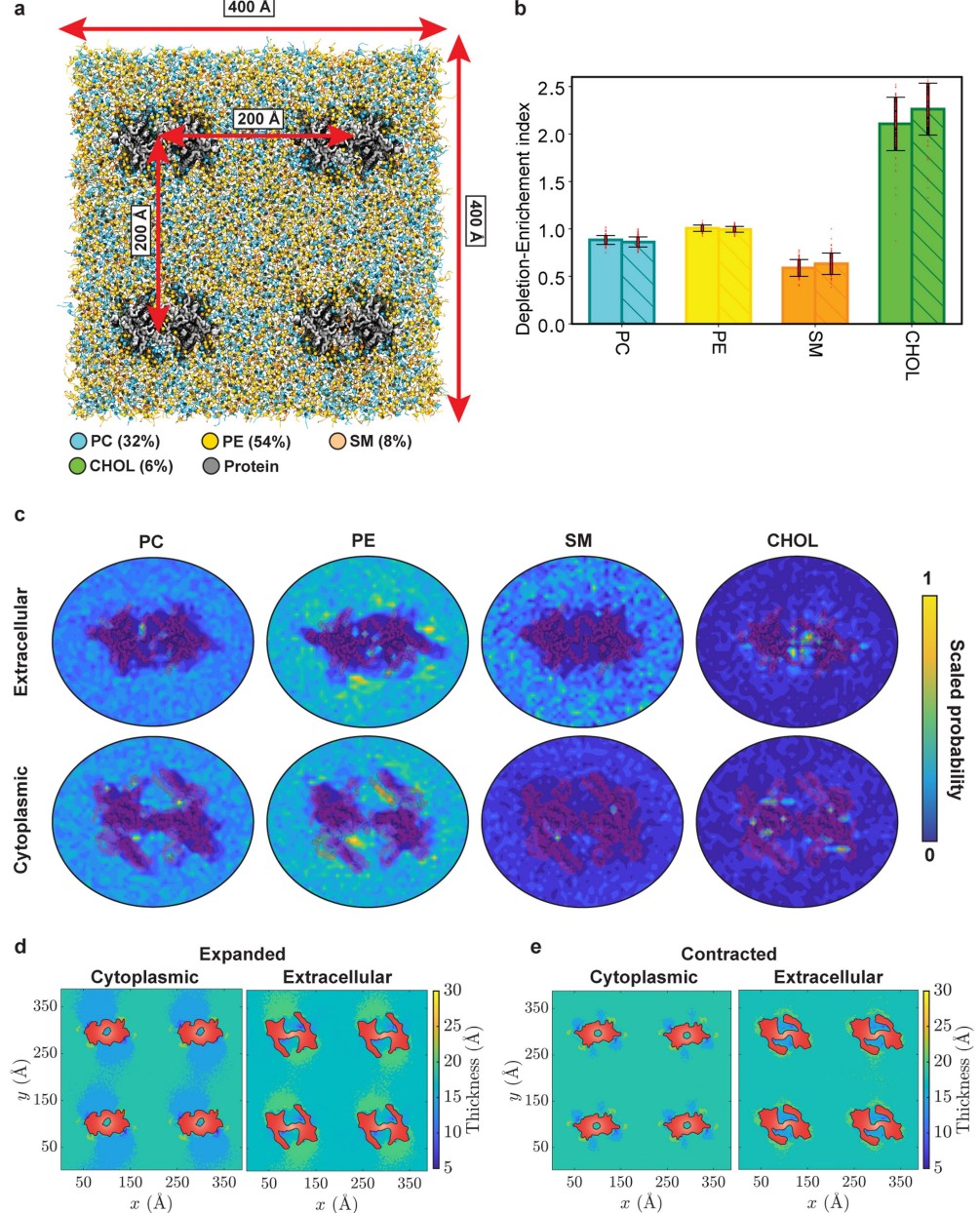

**Extended Data Fig. 10 | Coarse-grained MD simulations of TMC-1 complex in a membrane bilayer. a**, Four TMC-1 complexes (gray) in the E conformation embedded in a lipid bilayer composed of PC, PE, SM, and cholesterol (CHOL) shown in cyan, yellow, orange, and green, respectively, with a molar ratio of 32:54:8:6. **b**, Enrichment-depletion indexes of each lipid component in the proximity of the protein obtained from the simulations of the E (solid bars) and C (striped bars) conformations. PC and PE densities in the bulk and in proximity of the protein are similar, whereas SM is depleted and CHOL is enriched in the vicinity of the protein relative to their bulk concentrations. Data points are shown as red dots. Mean and error bars are calculated from 1000 data points in each bar plot and error bars are standard deviations of the data. **c**, Heatmaps representing distributions of different lipid species around the protein in the E conformation. Each distribution is calculated from the last 5 $\mu$s of the trajectory and averaged over all four protein copies. **d** and **e**, Thickness patterns calculated for the extracellular and cytoplasmic leaflets averaged over the last 5 $\mu$s of the CG trajectories of the E and C conformations, respectively. The cross-section of the protein is shown in red. The color scale represents the thickness of each leaflet, with blue and yellow corresponding to thinning and thickening, respectively. E and C conformations generate different membrane deformation patterns.

**Extended Data Table 1 | Statistics for 3D reconstruction and model refinement**

| States<br>Codes | Expanded<br>(EMD-26741)<br>(PDB **7USW**) | Contracted<br>(EMD-26742)<br>(PDB **7USX**) | With ARRD-6<br>(EMD-26743)<br>(PDB **7USY**) |
|---|---|---|---|
| **Data collection and processing** | | | |
| Microscope | | Titan Krios | |
| Camera | | K3 BioQuantum | |
| Magnification | | 105,000 | |
| Voltage (kV) | | 300 | |
| Defocus range (μm) | | -1.0 to -2.4 | |
| Exposure time (s) | | 3.329 | |
| Dose rate ($e^-$/Å$^2$/s) | | 14.9 | |
| Number of frames | | 50 | |
| Pixel size (Å) | | 0.831 (0.4195; Super-resolution) | |
| Micrographs (no.) | | 25,852 | |
| Initial particles (no.) | | 3,216,845 | |
| Symmetry imposed | | C2 | C1 |
| Final particles (no.) | 142,396 | 140,559 | 99,248 |
| Map resolution (Å) | 3.10 | 3.09 | 3.54 |
| FSC threshold | 0.143 | 0.143 | 0.143 |
| **Refinement** | | | |
| Initial model (PDB code) | De novo, Alphafold2, 6WUD | | Expanded |
| Model resolution (Å) | 3.27 | 3.24 | 3.89 |
| FSC threshold | 0.5 | 0.5 | 0.5 |
| Model composition | | | |
| Non-hydrogen atoms | 14,322 | 14,220 | 16,582 |
| Protein atoms | 13,552 | 13,544 | 16,364 |
| Ligand atoms | 770 | 676 | 218 |
| $B$ factors (Å$^2$) | | | |
| Protein | 46.17 | 55.26 | 65.34 |
| Ligand | 40.03 | 44.20 | 9.06 |
| R.m.s. deviations | | | |
| Bond length (Å) | 0.003 | 0.003 | 0.003 |
| Bond angle (°) | 0.511 | 0.546 | 0.649 |
| **Validation** | | | |
| Favored (%) | 96.87 | 96.25 | 97.01 |
| Allowed (%) | 3.13 | 3.75 | 2.99 |
| Disallowed (%) | 0 | 0 | 0 |
| Poor rotamers | 0 | 0 | 0 |
| MolProbity score | 1.59 | 1.53 | 1.64 |
| Clash score | 7.32 | 6.20 | 8.74 |

# Reporting Summary

## Statistics

For all statistical analyses, confirm that the following items are present in the figure legend, table legend, main text, or Methods section.

| n/a | Confirmed | |
|---|---|---|
| ☐ | ☒ | The exact sample size (*n*) for each experimental group/condition, given as a discrete number and unit of measurement |
| ☐ | ☒ | A statement on whether measurements were taken from distinct samples or whether the same sample was measured repeatedly |
| ☒ | ☐ | The statistical test(s) used AND whether they are one- or two-sided<br>*Only common tests should be described solely by name; describe more complex techniques in the Methods section.* |
| ☒ | ☐ | A description of all covariates tested |
| ☐ | ☒ | A description of any assumptions or corrections, such as tests of normality and adjustment for multiple comparisons |
| ☐ | ☒ | A full description of the statistical parameters including central tendency (e.g. means) or other basic estimates (e.g. regression coefficient) AND variation (e.g. standard deviation) or associated estimates of uncertainty (e.g. confidence intervals) |
| ☒ | ☐ | For null hypothesis testing, the test statistic (e.g. *F*, *t*, *r*) with confidence intervals, effect sizes, degrees of freedom and *P* value noted<br>*Give P values as exact values whenever suitable.* |
| ☒ | ☐ | For Bayesian analysis, information on the choice of priors and Markov chain Monte Carlo settings |
| ☒ | ☐ | For hierarchical and complex designs, identification of the appropriate level for tests and full reporting of outcomes |
| ☒ | ☐ | Estimates of effect sizes (e.g. Cohen's *d*, Pearson's *r*), indicating how they were calculated |

*Our web collection on statistics for biologists contains articles on many of the points above.*

## Software and code

Policy information about availability of computer code

| Data collection | SerialEM 3.8 |
|---|---|
| Data analysis | CryoSparc 3.3.1, PyMOL 2.4.1, Chimera 1.16, ChimeraX 1.1, MOLE 2.0, PHENIX 1.20, COOT 0.9, Alphafold2 (AF-D3KZG3-F1) |

For manuscripts utilizing custom algorithms or software that are central to the research but not yet described in published literature, software must be made available to editors and reviewers. We strongly encourage code deposition in a community repository (e.g. GitHub). See the Nature Portfolio guidelines for submitting code & software for further information.

## Data

Policy information about availability of data

All manuscripts must include a data availability statement. This statement should provide the following information, where applicable:
- Accession codes, unique identifiers, or web links for publicly available datasets
- A description of any restrictions on data availability
- For clinical datasets or third party data, please ensure that the statement adheres to our policy

WormBase gene code for TMC-1: T13G4.3.1. The coordinates and volumes for the cryo-EM data have been deposited in the Electron Microscopy Data Bank under accession codes EMD-26741 (Expanded), EMD-26742 (Contracted), and EMD-26743 (with ARRD-6). The coordinates have been deposited in the Protein Data Bank under accession codes 7USW (Expanded), 7USX (Contracted), and 7USY (with ARRD-6). All the initial and final snapshots of the MD trajectories, as well as simulation parameter and configuration files are deposited at https://doi.org/10.5281/zenodo.6780283.

# Field-specific reporting

Please select the one below that is the best fit for your research. If you are not sure, read the appropriate sections before making your selection.

☒ Life sciences        ☐ Behavioural & social sciences        ☐ Ecological, evolutionary & environmental sciences

For a reference copy of the document with all sections, see nature.com/documents/nr-reporting-summary-flat.pdf

# Life sciences study design

All studies must disclose on these points even when the disclosure is negative.

| | |
|---|---|
| Sample size | Sample sizes were not predetermined for this study. Sample sizes of cryo-EM data were determined by the availability of the microscope time. Single molecule pulldown experiments were carried out by analyzing 200 spots each from three movies. For spectral confocal imaging, ten worms of different larval and adult stages were imaged in two separate experiments. The sample sizes of these both experiments were determined based on the consistency and variability. |
| Data exclusions | The following exclusions were pre-established. Particles were removed if their 3D reconstructions had poor quality. |
| Replication | All experiments were performed with independent replicates as described and replicates were successful within expected variation. Cryo-EM related experiments including protein purification and SDS-PAGE gels were successfully reproduced at least three times independently. Spectral confocal images of ten worms of different larval and adult stages were collected in two separate experiments, on different days, and yielded identical results. Single molecule pulldown experiments were successfully replicated two times independently. |
| Randomization | Randomization is not relevant to the biochemical and structural experiments described in this work and would not impact the results of the experiments. |
| Blinding | The investigators were not blinded. Blinding is not technically or practically feasible for the biochemical and structural experiments described in this work and would not impact the interpretation of the results. |

# Reporting for specific materials, systems and methods

We require information from authors about some types of materials, experimental systems and methods used in many studies. Here, indicate whether each material, system or method listed is relevant to your study. If you are not sure if a list item applies to your research, read the appropriate section before selecting a response.

## Materials & experimental systems

| n/a | Involved in the study |
|---|---|
| ☐ | ☒ Antibodies |
| ☒ | ☐ Eukaryotic cell lines |
| ☒ | ☐ Palaeontology and archaeology |
| ☐ | ☒ Animals and other organisms |
| ☒ | ☐ Human research participants |
| ☒ | ☐ Clinical data |
| ☒ | ☐ Dual use research of concern |

## Methods

| n/a | Involved in the study |
|---|---|
| ☒ | ☐ ChIP-seq |
| ☒ | ☐ Flow cytometry |
| ☒ | ☐ MRI-based neuroimaging |

## Antibodies

| | |
|---|---|
| Antibodies used | Commercial antibodies: Anti-Flag M2 Affinity Gel (Sigma, A2220, lot:SLCJ7861), Anti-GFP nanobody: (https://www.addgene.org/browse/article/6869/). GFP nanobody used for single molecule pulldown experiments was diluted to at a concentration of 1 to 3 µg/mL. |
| Validation | Validation of the anti-GFP nanobody used for single molecule pulldown experiments can be found in the published literature (PMID: 20945358). Validation for the Anti-FLAG M2 affinity gel can be found on the manufacturers website (https://www.sigmaaldrich.com/US/en/product/sigma/a2220). |

## Animals and other organisms

Policy information about studies involving animals; ARRIVE guidelines recommended for reporting animal research

| | |
|---|---|
| Laboratory animals | Transgenic C. elegans were generated by SunyBiotech (strain: PHX2173 tmc-1(syb2173)). Worms of mixed larval and adult ages were grown and maintained in an incubator set at 20C. |
| Wild animals | The study did not involve wild animals. |

| Field-collected samples | No field-collected samples were used in this study. |
| Ethics oversight | C. elegans experiments do not require ethical approval. |

Note that full information on the approval of the study protocol must also be provided in the manuscript.

