## [Peer Review File · Nature]

Manuscript Title: Structure of *C. elegans* TMC-1 complex illuminates auditory mechanosensory transduction

Reviewer Comments & Author Rebuttals

Reviewer Reports on the Initial Version:

Referees' comments:

Referee #1 (Remarks to the Author):

Due to a wealth of difficulties regarding the study of the underlying molecular components, the molecular basis of mechanosensation remains among the last frontiers in sensory physiology. Perhaps nowhere is this challenge more acute than in the study of mechanotransduction in auditory systems. And while the physiology and genetics of vertebrate hair cells have been extensively studied for the better part of 4 decades, the molecular underpinnings of the mechanotransduction channel complex (MTC) has remained elusive. In this manuscript, Jeong, Clark, et al. take the first step towards filling that void. In a remarkable set of biochemical and structural biology experiments they have: 1- Achieved tagged expression of the native MTC in worms; 2- Carried out a biochemical and compositional (through mass spec) characterization of the complex; 3- Determined the cryoEM structure of the purified *C. elegans* MTC in a detergent micelle. They have done so, on the basis of remarkably limited sample material. This is a bona fide tour de force.

Among their key findings, Jeong, Clark, et al. report on the widely expected dimeric stoichiometry of the complex, the placement and interactions with associated partners (CALM1, ceTMIE, ARRD6), the deep penetration and interaction of lipids within the structure and the surprising absence of any ankyrin orthologs as part of the complex (expected on the basis of earlier work). Details on the analyses of the permeation pathway, putative influence of MTC on surrounding lipids, and the potential relevance of the expanded-compacted conformations to TMC function should be revised/expanded/reconsidered.

In the absence of a viable strategy to probe the functional aspects of this complex, the present contribution does a remarkable job at the general description of the complex and the overall interactions among its components, providing strong experimental support to a number of vertebrate MTC predictions, while opening the field to some unexpected results (what role for ARRD6?). All in all, the present work sets the stage for rapid advances in the understanding of the structure-function relations of TMC-like mechanotransduction complexes, offering a clear experimental alternative to the still daunting vertebrate MTC in auditory systems.

General comments

- It is understandable that the authors would try to highlight structural (and potentially functional) relationship between the *C. elegans* MTC and that of vertebrates (particularly mammals). I urge some care in this regard. While the overall sequence similarity profile with human TMC1 hovers around 30-35% (well within the threshold for fold similarity), some of the intriguing findings regarding the composition of the complex and the potential parallels regarding activation mechanisms should, at this stage be interpreted conservatively. The fact is, little mechanistic information exists regarding the process of mechanotransduction in the *C. elegans* MTC. In addition, while ceTMC1 appears associated with touch (and egg laying) it seems to localize in sensillum OLQ neurons (and other non-stereocilia containing cells). Perhaps the presence of a bound arrestin ortholog (and lack of the expected bound ankyrin) might be related to a non-equivalent mechanical activation. This point ought to be discussed further.

- Page 12, lines 274-278. Comments on the role of ionizable residues lining the pore on permeation properties of ceTMC1 too speculative. Eliminate or sharply tone down. No one has any idea of the residues that actually line the open conformation TMC1 permeation pathway.

- Page 13, "Expanded' and 'Contracted' conformations". The overall arguments highlighting the "expanded" and "contracted" TMC1 conformations appears to be too subtle to be mechanistically relevant for mechanotransduction. Indeed, observation of the putative permeation pathway in both conformers reveal little or no rearrangements (based on the provided maps). I have no doubt that these conformations are part of the structural landscape of TMC1 in a detergent micelle, but these might just be part of the conformational flexibility of the complex in the absence of external activating forces.

- Page 13-14 Membrane-MTC interactions. The computational analysis is expertly done, if somewhat limited. Proposals associated to the influence of the TMC on the surrounding lipid bilayer would need to be further developed and expanded. What is the actual role of the H3 helix on membrane distortion? What's the principle of the long range propagation of the membrane thinning effect? Is there any functional evidence that correlates with these computational results?

Specific comments

- TMIE in mammals is predicted to have two TM helices. Only TM1 has been conserved in *C. elegans*. Could TM2 be the helix that interacts with LHFPL5 and not TM1?

- The statement "The two helices located closest to inner leaflet, H3 and H4, are amphipathic, a common feature among mechanosensitive ion channels" should probably cite the original studies.

- Pan et al Neuron 2018 describes an extensive set of cysteine mutations which indirectly determines the residues facing the putative pore in mouse TMC1. In supplementary figure 3, it would be useful to align, map and contrast similarities and differences with the conserved residues on the *C. elegans* TMC1 structure.

- Given the technical achievement associated to this purification/structural determination pipeline, it would be informative to show a field of individual particles together with better images of the 2D classes.

Page 14, lines 331-335. The rationale for TMIE as transduction "handle" needs better development. If the main rationale is TMIE's proximity to the permeation pathway, it should be explicitly noted. I'm OK with speculative proposals, but the description of the putative link between the MTC and the LHFPL5/PCDH15 complex truly requires a cartoon of some sort...

Referee #2 (Remarks to the Author):

The present manuscript describes the cryoEM structure of the native mechanosensory transduction (MT) channel complex from *C. elegans*. This complex plays a pivotal role in the mechanotransduction pathway that underpins hearing and balance in mammals and light touch response in nematodes. The authors find that the MT complex is formed by the pore-forming TMC1 associated with three ancillary subunits, TMIE, CALM-1 and ARRD-6, the latter of which is present only in one protomer of a subset of the purified proteins. As predicted, the architecture of the transmembrane region of TMC1 is similar to that of the TMEM16 channels and scramblases and of the OSCA channels. The cytosolic region shows extended amphipathic helices that embed into the membrane and could play a role in mechanosensation. The single-pass TMIE subunit peripherally associates with TMC1 near the putative ion permeation pore. In contrast, the CALM-1 protein makes extensive interactions with the cytosolic domain of the TMC1 channel and with the arrestin-

like ARR-6.

Overall, this is an excellent manuscript. These novel structures provide the first insights into the molecular architecture of the MT complex and will be of interest to a broad readership. The structures are of high quality, their description clear and concise. I have no major concerns, only minor suggestions to expand the discussion and improve the presentation.

1) The authors claim that the cavity between TMIE and TMC1 contains at least 8 lipid molecules. However, the deposited PDB files show only 1-2 lipids per cavity and several small partial densities that could be attributed to detergent molecules from the micelles. The MD data in Extended Fig. 9 only shows that there is a relative enrichment of cholesterol in this cavity but does not provide insights into the number of lipid molecules. This should be toned down or documented better.

2) The analysis of the TMC1 pore could be improved.

a. Is the position of the three constrictions in the pore conserved in the TMEM16A and OSCA channels? Recent work from the Dutzler lab suggested that TMEM16A is gated at by a hydrophobic constriction near the center of the permeation pathway (Lam et al., Nat Comms, 2021). Do the positions of these hydrophobic constrictions (or of the other constrictions) align? This should be discussed.

b. I find the plot of the permeation pathway in Fig. 5c and associated description in page 12 somewhat misleading. The current description suggests this channel forms a ~100 Å long pore. This is more than twice the length of a typical membrane and a quick inspection of the PDBs shows this is not the case. I guess this is due to the winding, non-linear path identified by the MOLE program. Wouldn't it be more informative to plot the radius as a function of the z-coordinate relative to the plane of the membrane? If this is not possible, this point should be clarified.

c. The effects on ion conduction and selectivity of multiple mutants localizing to the putative TMC1 pore has been described by several groups. It would be important to discuss these results in the context of the current structures and add a figure panel showing the position of these residues.

3) The authors describe 2 conformations of the channel complex dubbed as Contracted and Expanded. In what conformation is the protein when in complex with the ARR-6 molecule? Looking at the PDB files it seems to align well to the E conformation. This should be stated.

4) Why were the MD simulations ran only using the E conformation? If the membrane deformation has mechanistic implications for mechanosensation, as suggested by the authors, then it is important to determine whether the E and the C conformations display similar deformation patterns.

5) Do the authors observe interprotomer dynamics during the MD simulations that could suggest a rearrangement from the E to the C state?

6) The CG analysis of the long-range effects on membrane deformation is interesting. However, I wonder whether the high density of TMC1 channels used in the computational experiments represents a physiologically meaningful condition, especially given the low abundance of the channel in native tissues. This should be discussed.

Minor points

- Fig. 2c, please label H4, the text refers to this helix and I could not follow it just looking at the figure, but had to use the PDB to orient myself.

- The PDB models do not show connections at the palmitoylation sites for C44 of TMIE. For example, in 7USW the lipid and C44 are not connected, whereas the density at that location is

clearly continuous and the model shown in Fig. 3d also shows the covalent linkage.

- Fig. 3a is confusing, the arrow seems to indicate the highlighted area between TM6-7-8 as the pore, whereas the text and MOLE analysis clearly state that the putative pore is between TM4-5-6-7, in good analogy to what is seen in the TMEM16A and OSCA channels.

- The authors should mention that the structural similarity of TMC1 with the TMEM16 proteins was predicted by several groups and add relevant citations (i.e. Ballesteros et al., *Elife*, 2018; Pan et al., *Neuron*, 2018; Walujkar et al., *bioRxiv*, 2021)

Referee #3 (Remarks to the Author):

This work presents the (Cryo-EM) structure of the TMC-1 complex that is associated with mechanosensory signal transduction. In mammals this is thought to underpin hearing and balance pathways. The structure represents a significant step-forward in our understanding of these mechanisms and represents an original and significant piece of work. I congratulate the authors on the effort taken to obtain these structures. In general, the manuscript is well written, although it mainly is simply a description of the observed structure with some MD used to try and support the observations. There is agreement with some previous mutagenesis and mutations from the literature, but I am not an expert on that so cannot comment directly on how well it supports all previous observations.

I do however have the following comments and suggestions for improvements:

1. The structure reveals an unexpectedly large "cavity" presumably filled with lipids in between the TMC-1 protein and the TMIE protein. Given the unusual nature of this and the fact that the structure has been solved in detergent, it would be useful to have more guidance as to whether this is simply artefactual from being in detergent. This is especially problematic for a complex that is likely to be very sensitive to the lipid conditions. Furthermore, is the sensitivity to lipid actually known?

2. To some extent the authors try to explore this a little bit with the MD approach. However, there are a number of issues that need addressing:-

a) What was the reasoning behind the choice of lipids? And the choice of ratio? Sphingomyelin is a generic term – what is the exact chain composition?

b) The coarse-grained approach seems slightly flawed. Although the authors set up 4 proteins in the system, they appear to be set up in the exactly the same orientation and thus there is a clear period influence effect between each of them. These simulations should be repeated with different starting orientations in each of the four protein complexes. The statement of page 14 (line 321 onwards) is in my view likely to simply reflect this artificial periodicity.

c) How well is the lipid density in Figure 3d captured by the atomistic simulations? Do the simulated lipids overlay with that lipid?

d) To what degree do the coarse-grain and MD "thinning" profiles agree? Can this be quantified? It is almost impossible to tell and what does black mean in Extended Figure 9d?

e) As I understand it, protein backbone restraints were present throughout the atomistic simulations. Do the authors see that the structure is unstable without these restraints? I realise

the aim was to perhaps look at lipid interactions, but it would be extremely useful to know if these coordinates are stable without restraints (and hence would also help inform as to the relevance of this structure in detergent)

f) What is the interpretation of the H3 helices in the atomistic simulations going deeper into the membrane – is this simply a consequence of taking the protein out of detergent and placing into a bilayer? The authors speculate that H3 acts like a float to open the channel. It's difficult to see how this would work – is it not possible that H3 actually provides a fulcrum to lever against?

g) I assume Propka did not reveal any unusual pKas?

Finally, please also deposit at least final snapshots and input (including parameter) files of the simulation data for CG and atomistic runs to help ensure reproducibility. I view this as important and non-negotiable.

Author Rebuttals to Initial Comments:

Referees' comments:

Referee #1 (Remarks to the Author):

General comments

- It is understandable that the authors would try to highlight structural (and potentially functional) relationship between the C. elegans MTC and that of vertebrates (particularly mammals). I urge some care in this regard. While the overall sequence similarity profile with human TMC1 hovers around 30-35% (well within the threshold for fold similarity), some of the intriguing findings regarding the composition of the complex and the potential parallels regarding activation mechanisms should, at this stage be interpreted conservatively. The fact is, little mechanistic information exists regarding the process of mechanotransduction in the C. elegans MTC. In addition, while ceTMC1 appears associated with touch (and egg laying) it seems to localize in sensillum OLQ neurons (and other non-stereocilia containing cells). Perhaps the presence of a bound arrestin ortholog (and lack of the expected bound ankyrin) might be related to a non-equivalent mechanical activation. This point ought to be discussed further.

Reply. We thank the reviewer for this comment and agree that it is important to be conservative when comparing worm and vertebrate TMC-1 complexes due to functional differences in their respective organisms. We have therefore toned-down the mechanistic comparisons between the two orthologs throughout the text, leaving the majority of the discussion for the summary paragraph, in which we speculate how the TMC-1 complex may interact with PCDH15 to sense force in vertebrates.

- Page 12, lines 274-278. Comments on the role of ionizable residues lining the pore on permeation properties of ceTMC1 too speculative. Eliminate or sharply tone down. No one has any idea of the residues that actually line the open conformation TMC1 permeation pathway.

Reply. We agree that the residues lining the open conformation of the pore might be different than those observed in the closed state. We have emphasized this point in the text by editing the following sentence at line 283:

“While the ionizable residues lining the closed pore are predominately basic and thus not in keeping with a canonical Ca²⁺-permeable channel, which typically harbors acidic residues, we note that the residues lining the permeation pathway in its open conformation are presently unknown.”

- Page 13, “Expanded’ and ‘Contracted’ conformations”. The overall arguments highlighting the “expanded” and “contracted” TMC1 conformations appears to be too subtle to be mechanistically relevant for mechanotransduction. Indeed, observation of the putative permeation pathway in both conformers reveal little or no rearrangements (based on the provided maps). I have no doubt that these conformations are part of the structural landscape of TMC1 in a detergent micelle, but these might just be part of the conformational flexibility of the complex in the absence of external activating forces.

Reply. We agree that the conformational changes between 'E' and 'C' are too subtle to be mechanically relevant for mechanotransduction. Thus, we have moved Fig. 6a, which shows the 'E' and 'C' conformations, to Extended Data Fig. 8 and have toned down the associated text. We do note, nevertheless, that in the new simulations of the 'C' conformation, we observe different patterns of membrane deformation when compared to the 'E' conformation (Extended

Data Fig. 9e and Supplementary Fig. 5), suggesting that transition between the 'E' and 'C' conformations can affect the membrane mechanical properties.

- Page 13-14 Membrane-MTC interactions. The computational analysis is expertly done, if somewhat limited. Proposals associated to the influence of the TMC on the surrounding lipid bilayer would need to be further developed and expanded. What is the actual role of the H3 helix on membrane distortion? What's the principle of the long range propagation of the membrane thinning effect? Is there any functional evidence that correlates with these computational results?

Reply. We are entirely onboard with these comments yet are also constrained by space, thus limiting the extent to which we can respond, as fully as the reviewer suggests, to all of the comments. Nevertheless, we begin by noting that the level of lipid perturbations at the protein interface determines the degree of its propagation into the membrane. In the case of TMC-1 in the 'E' conformation, substantial membrane distortion in the proximity of the protein generates a long-range membrane deformation pattern across the bilayer. Here, the deep partitioning of H3 into the membrane, which is highlighted in Fig. 5a-b, as well as other membrane-proximal structural elements in the protein, appear to be major contributors to the observed membrane deformation (Fig. 5c and Extended Data Fig. 9d). While the role(s) of H3 in mechanotransduction of TMC-1 are yet to be elucidated, we note that in the new simulations of the 'C' conformation, we observe different patterns of membrane deformation when compared to the 'E' conformation (Extended Data Fig. 9e and Supplementary Fig. 5), suggesting that transition between the 'E' and 'C' conformations can affect the membrane mechanical properties. A short discussion on this effect is added to the revised manuscript at line 332:

"Interestingly, the 'E' and 'C' conformations generate distinct membrane deformation patterns (Fig. 5c, Extended Data Fig. 9d-e, and Supplementary Fig. 5). In particular, the 'E' conformation induces a more pronounced membrane distortion (thickening/thinning) close to the protein, resulting in a longer-range membrane deformation propagation, as much as ~50 Å away from the protein (Extended Data Fig. 9d-e)."

Specific comments

- TMIE in mammals is predicted to have two TM helices. Only TM1 has been conserved in *C. elegans*. Could TM2 be the helix that interacts with LHFPL5 and not TM1?

Reply. As discussed in Supplementary Fig 2, we present data that is consistent with the conclusion that the first transmembrane domain of TMIE is a signal peptide and that following processing by the signal peptidase, residues 1-28 of mouse TMIE, are no longer covalently attached to the remainder of TMIE. We thus conclude that mature TMIE, in mammals, has a single TM domain, referred to as TM1. An alignment of protein sequences from mouse, human, zebrafish and *C. elegans* indicates TM1 is conserved. As shown in the structure in Fig. 2, TMIE consists of a single transmembrane domain followed by an 'elbow-like' linker and a cytosolic helix. The interaction between TMIE and TMC-1 is mediated primarily by the cytosolic TMIE elbow.

- The statement "The two helices located closest to inner leaflet, H3 and H4, are amphipathic, a common feature among mechanosensitive ion channels" should probably cite the original studies.

Reply. We agree that it is usually best to cite primary studies. In this case, however, we believe it is appropriate to cite the review. The review by Kefauver et al. describes how amphipathic helices are common features of mechanosensitive ion channels and highlights amphipathic helices in the structures of six different mechanically activated channels. Several of the original studies, while presenting the structures, do not actually discuss the amphipathic nature of these helices. We therefore suggest that, in this instance, it is more helpful to the reader to cite the review.

- Pan et al *Neuron* 2018 describes an extensive set of cysteine mutations which indirectly determines the residues facing the putative pore in mouse TMC1. In supplementary figure 3, it would be useful to align, map and contrast similarities and differences with the conserved residues on the *C. elegans* TMC1 structure.

Reply. We agree that the manuscript would benefit from discussion of the cysteine mutagenesis results from Pan et al. as well as from the recently published cysteine mutagenesis results from Akyuz et al. We think that this data fits well in Extended Data Figure 7, which shows the locations of deafness and hearing loss mutations on the *C. elegans* TMC-1 complex structure. We have added a panel to Extended Data Figure 7 in which we show the locations of cysteine mutants that alter pore properties of mouse TMC-1. We have also added the following discussion to line 276 of the text:

“Cysteine mutagenesis experiments have identified eight pore-lining residues in mouse TMC-1 and while five of these residues are not conserved in *C. elegans*, the locations of all eight map to putative pore-lining positions in *C. elegans* TMC-1 (Extended Data Fig. 7a), thus supporting their location in the ion permeation pathway.”

Revised Extended Data Figure 7:

- Given the technical achievement associated to this purification/structural determination pipeline, it would be informative to show a field of individual particles together with better images of the 2D classes.

Reply. We appreciate this comment and have revised Extended Data Fig. 5a to show particles on a representative micrograph together with several 2D classes of each major 3D class. Each different colored circle (green, red and blue) on the micrograph indicates particles that were subsequently classified to the Expanded, Contracted and with ARR-6 bound conformations, respectively.

Revised Extended Data Fig. 5a.

Page 14, lines 331-335. The rationale for TMIE as transduction “handle” needs better development. If the main rationale is TMIE’s proximity to the permeation pathway, it should be explicitly noted. I’m OK with speculative proposals, but the description of the putative link between the MTC and the LHFPL5/PCDH15 complex truly requires a cartoon of some sort...

Reply. The reviewer is correct, we speculate that TMIE acts as a transduction handle due to its proximity to the putative ion permeation pathway and its interaction with TMC-1 pore-forming helices. TMIE forms hydrogen bonds with TMC-1, TM6 and TM8 and the acyl chain of the palmitoylated TMIE C44 makes extensive contacts with TM8. We have added the following text to the summary paragraph to more clearly describe our rationale:

“TMIE is positioned proximal to the putative pore-forming helices TM4-8 (Fig. 2a), interacting directly with TMC-1, TM6 and TM8 through hydrogen bonds and via the palmitoyl group attached to TMIE residue C44. The arrangement of subunits resembles an accordion, with TMIE forming the instrument ‘handles’. We speculate that force applied to TMIE, either via the membrane or by way of direct contacts with an auxiliary subunit, is then coupled to the TMC-1 pore forming TMs, opening the pore to allow ion permeation.”

Additionally, we incorporated a speculative cartoon in Figure 5d that shows how, in vertebrates, PCDH15/LHFPL5 may interact with the TMC-1 complex and participate in mechanotransduction.

Referee #2 (Remarks to the Author):

1) The authors claim that the cavity between TMIE and TMC1 contains at least 8 lipid molecules. However, the deposited PDB files show only 1-2 lipids per cavity and several small partial densities that could be attributed to detergent molecules from the micelles. The MD data in Extended Fig. 9 only shows that there is a relative enrichment of cholesterol in this cavity but does not provide insights into the number of lipid molecules. This should be toned down or documented better.

Reply. We appreciate this comment and have edited the sentence to tone down the text. (line 181) “~ that is occupied by at least eight lipid molecules.” to “~ that is occupied by multiple detergent or lipid molecules.”

2) The analysis of the TMC1 pore could be improved.

a. Is the position of the three constrictions in the pore conserved in the TMEM16A and OSCA

channels? Recent work from the Dutzler lab suggested that TMEM16A is gated at by a hydrophobic constriction near the center of the permeation pathway (Lam et al., Nat Comms, 2021). Do the positions of these hydrophobic constrictions (or of the other constrictions) align? This should be discussed.

Reply. We appreciate this comment and have investigated whether the locations of the TMC-1 constriction sites are conserved in the evolutionarily related OSCA1.2 and TMEM16a ion channels. Superposition of TMC-1 with the OSCA1.2 (PDB 6MGV) and TMEM16a (PDB 5OYB) structures reveals that the position and composition of the second constriction site is conserved in OSCA1.2 and TMEM16a. The neck region of OSCA1.2 is primarily composed of hydrophobic residues and aligns well with the location of the second construction site, as does the TMEM16a hydrophobic gate. We have added the following sentence at line 272 of the manuscript:

“Interestingly, the location of the second constriction site aligns with the narrow ‘neck’ region of the putative OSCA1.2 pore, which is lined by primarily hydrophobic residues, as well as with the hydrophobic TMEM16a gate (Lam et al., Nat. Comms., 2021; Jojoa Cruz et al., eLife, 2018).”

b. I find the plot of the permeation pathway in Fig. 5c and associated description in page 12 somewhat misleading. The current description suggests this channel forms a ~100 Å long pore. This is more than twice the length of a typical membrane and a quick inspection of the PDBs shows this is not the case. I guess this is due to the winding, non-linear path identified by the MOLE program. Wouldn't it be more informative to plot the radius as a function of the z-coordinate relative to the plane of the membrane? If this is not possible, this point should be clarified.

Reply. We have altered the permeation plot in Figure 4c (previously Figure 5c) to show the Z-coordinate on the Y-axis instead of the ‘Distance along pore axis’.

Revised Figure 4:

c. The effects on ion conduction and selectivity of multiple mutants localizing to the putative TMC1 pore has been described by several groups. It would be important to discuss these results in the context of the current structures and add a figure panel showing the position of these residues.

Reply. We appreciate this suggestion and have added a panel to Extended Data Figure 7 that shows the locations of pore lining residues, as identified by cysteine mutagenesis experiments. Please also see our reply to a similar comment from Reviewer #1 for a more complete response.

3) The authors describe 2 conformations of the channel complex dubbed as Contracted and Expanded. In what conformation is the protein when in complex with the ARRD6 molecule? Looking at the PBD files it seems to align well to the E conformation. This should be stated.

Reply. We sincerely appreciate this comment and have defined the conformation of ARRD-6 complex.

(line 242) “Structural alignment shows the core complex in ARRD-6 complex is most similar to the ‘E’ conformation.”

4) Why were the MD simulations ran only using the E conformation? If the membrane deformation has mechanistic implications for mechanosensation, as suggested by the authors, then it is important to determine whether the E and the C conformations display similar deformation patterns.

Reply. We thank the reviewer for this helpful suggestion. We have added to the revised manuscript simulations of the ‘C’ conformation both at all-atom (AA) and coarse-grained (CG) levels. At the AA level, three independent replicas of the ‘C’ conformation were simulated for 1 μ s each, and at the CG level a system containing four TMC-1 complexes embedded into a larger patch of membrane was simulated for 10 μ s. In the new simulations of the ‘C’ conformation, we capture different membrane deformation patterns (Extended Data Fig. 9e and Supplementary. Fig. 5) when compared to the ‘E’ conformation, which are indicative of the impact of the protein conformational changes on the mechanics of the membrane. Different arrangements of the H1-H3 helices in ‘E’ and ‘C’ conformations appear to be the main contributors to the observed difference.

5) Do the authors observe interprotomer dynamics during the MD simulations that could suggest a rearrangement from the E to the C state?

Reply. The MD simulations were performed primarily to study the effect of the protein, at its experimentally captured conformation(s), on the membrane. As such, in all simulations (both CG and AA) restraints were used to maintain the protein backbone close to the cryo-EM obtained structure(s), which prevent any possible transitions between the ‘E’ and ‘C’ conformations.

6) The CG analysis of the long-range effects on membrane deformation is interesting. However, I wonder whether the high density of TMC1 channels used in the computational experiments represents a physiologically meaningful condition, especially given the low abundance of the channel in native tissues. This should be discussed.

Reply. The four TMC-1 complexes incorporated in our CG simulations were designed to be far enough apart to represent four independent protein replicas. To confirm that this arrangement did not affect the observed membrane deformation patterns and propagation, we simulated a control system containing four TMC-1 in random (not aligned) orientations. As shown in the figure below, the membrane deformation patterns around each TMC-1 complex are nearly identical in the new simulations, verifying the independence of the four protein copies. While we

do note that the overall number of TMC-1 complexes in stereocilia is small, there is recent evidence for multiple TMC-1 complexes in stereocilia (Beurg et al., 2018).

Minor points

- Fig. 2c, please label H4, the text refers to this helix and I could not follow it just looking at the figure, but had to use the PDB to orient myself.

Reply. We appreciate this comment and have revised Fig. 1c to include a label for the H4 helix.

- The PDB models do not show connections at the palmytoilation sites for C44 of TMIE. For example, in 7USW the lipid and C44 are not connected, whereas the density at that location is clearly continuous and the model shown in Fig. 3d also shows the covalent linkage.

Reply. We have revised the PDB models, making a connection between Cys44 and the lipid.

- Fig. 3a is confusing, the arrow seems to indicate the highlighted area between TM6-7-8 as the pore, whereas the text and MOLE analysis clearly state that the putative pore is between TM4-5-6-7, in good analogy to what is seen in the TMEM16A and OSCA channels.

Reply. We appreciate this comment and have revised Fig. 3a (now Fig. 2a), indicating the highlighted area between TM4-8, accordingly.

- The authors should mention that the structural similarity of TMC1 with the TMEM16 proteins was predicted by several groups and add relevant citations (i.e. Ballesteros et al., *Elife*, 2018; Pan et al., *Neuron*, 2018; Walujkar et al., *bioRxiv*, 2021)

Reply. We appreciate this comment and have added a brief discussion of predicted TMC-1 structures, along with the relevant citations.

Referee #3 (Remarks to the Author):

1. The structure reveals an unexpectedly large “cavity” presumably filled with lipids in between the TMC-1 protein and the TMIE protein. Given the unusual nature of this and the fact that the structure has been solved in detergent, it would be useful to have more guidance as to whether this is simply artefactual from being in detergent. This is especially problematic for a complex that is likely to be very sensitive to the lipid conditions. Furthermore, is the sensitivity to lipid actually known?

Reply. We appreciate this comment. We do not think that the cavity between TMIE and TMC-1 is an artifact of detergent purification because the TMIE residues that interact with TMC-1 are located at the extracellular and intracellular boundaries of the TMIE transmembrane domain (W25 and I48-R52). These TMIE residues are highly conserved (Fig. 2e), suggesting that they are critical for the TMIE/TMC-1 interaction. We did not observe direct interaction between the TMIE and TMC-1 transmembrane helices, but rather lipid mediated interactions in the form of TMIE C44 palmitoylation, as well as many lipid or detergent molecules. In a plasma membrane, we hypothesize that this cavity would be filled with lipids that functionally bridge the TMIE and TMC-1 transmembrane domains. The MT complex in vertebrates is sensitive to lipid

composition and TMIE has been shown to bind to PIP₂ in its C-terminal domain, influencing MT channel conductance and ion selectivity in mice (Cunningham et al., Neuron, 2020). Other studies of vertebrate MT corroborate this result, demonstrating that PIP₂ additionally influences MT channel resting open probability and adaptation (Effertz et al., J. Neurosci., 2017; Hirono et al., Neuron, 2004). For clarification, we have added the following sentence to line 343 of the text:

“Interactions between TMIE and TMC-1 are restricted to the extracellular and intracellular boundaries of TMIE TM1, resulting in a large intramembranous cavity between the two subunits that we hypothesize is filled with lipid molecules in a plasma membrane environment.”

2. To some extent the authors try to explore this a little bit with the MD approach. However, there are a number of issues that need addressing:-

a) What was the reasoning behind the choice of lipids? And the choice of ratio? Sphingomyelin is a generic term – what is the exact chain composition?

Reply. The lipid composition was based on the plasma membrane composition of *C. elegans*, described in “Phospholipids from the free-living nematode *Caenorhabditis elegans*”, Satouchi, et al., Lipids, 28: 837-40 (1993) - <https://doi.org/10.1007/BF02536239>.

The sphingomyelin used in the CG simulations is the DPSM lipid in Martini lingo, corresponding to C(d18:1/18:0) N-stearoyl-D-erythro tails. In the AA simulations, sphingomyelin is PSM in CHARMM forcefield corresponding to N-palmitoyl-sphingomyelin (PSM). This information is now added to the Methods section.

b) The coarse-grained approach seems slightly flawed. Although the authors set up 4 proteins in the system, they appear to be set up in the exactly the same orientation and thus there is a clear period influence effect between each of them. These simulations should be repeated with different starting orientations in each of the four protein complexes. The statement of page 14 (line 321 onwards) is in my view likely to simply reflect this artificial periodicity.

Reply. To verify that the used arrangement has not influenced our conclusions regarding the deformation of the membrane around individual proteins, and as suggested by the reviewer, we have set up and performed a control simulation in which the four TMC-1 complexes are placed in random orientations. As shown in the figure below, the membrane deformation patterns are nearly identical to what we observed in our original set of simulation.

c) How well is the lipid density in Figure 3d captured by the atomistic simulations? Do the simulated lipids overlay with that lipid? d) To what degree do the coarse-grain and MD “thinning” profiles agree? Can this be quantified? It is almost impossible to tell and what does black mean in Extended Figure 9d?

Reply. As the reviewer is likely aware, the limited timescale of the atomistic simulations prevents one from obtaining sufficient sampling of the lipid distributions (limited by their slow lateral diffusion and mixing) around the protein. As such, the distribution of lipids around the protein is largely dictated by their initial configurations in atomistic simulations (in our case, the final configuration from the CG simulations). However, one can expect the deformation patterns around the protein to be sufficiently captured by the atomistic simulations. As shown in the figure below, the membrane deformation patterns between atomistic (AA) and CG simulations are quite consistent. Notably, the deformation pattern from AA simulations of the ‘E’ conformation was already reported in the main text (Fig. 5c), and we have added a new figure reporting the deformation pattern observed in AA simulations of the ‘C’ conformation (Supplementary Fig. 5).

The confusing black color was done by Matlab to outline different regions/levels of the heatmap. The outlines are now removed (Fig. 5c and Extended Data Fig. 9d-e).

e) As I understand it, protein backbone restraints were present throughout the atomistic simulations. Do the authors see that the structure is unstable without these restraints? I realise the aim was to perhaps look at lipid interactions, but it would be extremely useful to know if these coordinates are stable without restraints (and hence would also help inform as to the relevance of this structure in detergent)

Reply. As the reviewer correctly mentions, the main purpose of all the simulations was to characterize the behavior and structure of the lipid bilayer in response to the protein at experimental structures. To follow the reviewer’s comment, in a new set of atomistic simulations, three independent copies of TMC-1 complex were each simulated for 1 μ s, without applying any positional restraints. The calculated backbone RMSD of different domains (see Figure below), with respect to the cryo-EM structure for all the three replicas, shows the relative stability of the TMC-1 and CALM-1 domains. In contrast, the transmembrane TMIE helix, which resides at the periphery of the complex in both structures, fluctuates much more (RMSD reaching values more than 12 Å). Whether these large fluctuations are related to different lipidic environments of experiment (detergent) and simulations, or due to their different temperatures, requires additional studies.

The figure below shows the RMSD values of different structural elements of the protein in all three 1- μ s simulation replicas, as well as the initial and final configurations of the complex. In

the RMSD plots, colors correspond to different domains in TMC-1 complex, highlighted in the right panels.

f) *What is the interpretation of the H3 helices in the atomistic simulations going deeper into the membrane – is this simply a consequence of taking the protein out of detergent and placing into a bilayer? The authors speculate that H3 acts like a float to open the channel. It's difficult to see how this would work – is it not possible that H3 actually provides a fulcrum to lever against?*

Reply. We appreciate these comments. First, the mention of a ‘deeper’ penetration of H3 into the membrane was in reference to other membrane proteins simulated in the lab. To avoid confusion, we have replaced “deeper” with “deep” in the text. Second, we are well aware that the role of H3 in the mechanism of the TMC-1 complex is yet to be determined. Nevertheless, we suggest that H3 acts as a float rather than a fulcrum for two reasons. First, H3 extends away from the TMC-1 transmembrane domains and is not ‘anchored’ in place by direct interaction with CALM-1 or other domains of TMC-1. It therefore appears to be ‘floating’ untethered in the membrane, like a buoy. To act as a fulcrum, H3 would need to be securely positioned in the membrane to allow the transmembrane domains to lever against it. Second, H3 is connected to TMC-1 TM1, which in turn interacts with the putative pore-forming helix TM8. We envision a scenario wherein the membrane thins due to stretch or some other force, causing the vertical position of H3 to change with respect to the membrane plane and altering the position of TM1 and TM8.

g) *I assume Propka did not reveal any unusual pKas?*

Reply. Correct. We have added a short sentence to Methods to highlight this.

Finally, please also deposit at least final snapshots and input (including parameter) files of the simulation data for CG and atomistic runs to help ensure reproducibility. I view this as important and non-negotiable.

Reply. All the initial and final snapshots, as well as simulation parameter and configuration files are deposited: <https://doi.org/10.5281/zenodo.6780283>. The link is also added to the manuscript.

Reviewer Reports on the First Revision:

Referees' comments:

Referee #1 (Remarks to the Author):

The authors have done an excellent job responding to my queries. As far as this reviewer is concerned, this manuscript is ready for publication.

Referee #2 (Remarks to the Author):

The authors have addressed most of my concerns. I have a couple of small requests/suggestions regarding the additional MD results:

1) The difference in membrane deformation pattern between the E and C conformations is quite striking. Do the simulations reveal the structural basis for this difference? It is interesting that a relatively subtle rearrangement in the protein conformation results in such a different behavior. Could this difference have implications for mechanosensation by TMC1?

2) Are there differences in the lipid enrichment profiles between the E and C conformations? If not, this should be stated. If yes, this should be documented and discussed.

Referee #3 (Remarks to the Author):

The authors have done a great job at addressing my concerns (and indeed all concerns). I appreciate the extra simulation data and extensive revision of figures that have greatly improved the clarity of this work. I have no hesitation in recommending publication.

Author Rebuttals to First Revision:

Replies to Reviewer 2

Comment. 1) *The difference in membrane deformation pattern between the E and C conformations is quite striking. Do the simulations reveal the structural basis for this difference? It is interesting that a relatively subtle rearrangement in the protein conformation results in such a different behavior. Could this difference have implications for mechanosensation by TMC1?*

Reply. We appreciate these points. At this juncture, we speculate, and describe in the text, that the relative differences in conformation of H3 may largely underpin the differences in membrane deformation of the E and C conformations. As we state in the text, however, further studies are required.

We also do not know how these differences in protein conformation and membrane deformation impact mechanosensation. These important points, as well, require further investigation that is simply beyond the scope of the present study.

To address the comments of the reviewer, nevertheless, we have revised elements of the main text as follows (starting at line 319):

“As described in the next section, these changes are accompanied by significant alterations in the structure and mechanical behavior of the membrane surrounding the protein (Extended Data Fig. 9). Further studies, however, will be necessary to determine their ramifications on the function of the MT complex.”

Comment 2) *Are there differences in the lipid enrichment profiles between the E and C conformations? If not, this should be stated. If yes, this should be documented and discussed.*

Reply. We repeated the calculation of lipid enhancement/depletion for the C conformation and updated Panel b of Extended Data Figure 9 to include both data sets.

We see no meaningful differences in the lipid enrichment profiles between the E and C conformations. We have thus included, in the legend of Extended Data Fig. 9b, the following caption.

“b, Enrichment-depletion indexes of each lipid component in the proximity of the protein obtained from the simulations of the ‘E’ (solid bars) and ‘C’ (striped bars) conformations are similar. PC and PE densities in the bulk and in proximity of the protein are similar, whereas SM is depleted and CHOL is enriched in the vicinity of the protein relative to their bulk concentrations.